# Trauma-affected refugees treated with basic body awareness therapy or mixed physical activity as augmentation to treatment as usual —A pragmatic randomised controlled trial

Maja Sticker Nordbrandt[1,2], Charlotte Sonne[1], Erik Lykke Mortensen[3], Jessica Carlsson[1,2]*

**1** Competence Centre for Transcultural Psychiatry, Mental Health Centre Ballerup, Mental Health Services of the Capital Region, Ballerup, Denmark, **2** Faculty of Health and Medical Sciences, University of Copenhagen, Copenhagen, Denmark, **3** Department of Public Health and Center for Healthy Aging, University of Copenhagen, Copenhagen, Denmark

* jessica.carlsson@regionh.dk

## Abstract

### Background

The prevalence of post-traumatic stress disorder (PTSD) is estimated to be as high as 30% among refugees. The coexistence of prevalent chronic pain is believed to maintain symptoms of PTSD and add complexity to the condition. Despite this, little evidence exists on how to treat PTSD and comorbid conditions best in trauma-affected refugees.

### Aim

The aim of the present study was to investigate if adding either BBAT or mixed physical activity to the treatment as usual (TAU) for trauma-affected refugees with PTSD would increase the treatment effect compared to TAU alone.

### Method

Randomised controlled trial, 3-armed parallel group superiority study, conducted at Competence Centre for Transcultural Psychiatry, Denmark. Participants were adult trauma-affected refugees with PTSD. Allocation ratio was 1:1:1, stratified for PTSD severity and gender. An open-label design was applied due to the nature of the intervention. Participants were randomised to receive either individual basic body awareness therapy (group B) or individual mixed physical activity (group M) one hour/week for 20 weeks plus TAU, or TAU only (group C). The primary outcome was PTSD severity measured by Harvard Trauma Questionnaire (HTQ). Trial registration: ClinicalTrials.gov, NCT01955538.

### Results

Of the 338 patients included (C/B/M = 110/114/114), 318 patients were eligible for intention-to-treat analysis (C/B/M = 104/105/109). On the primary outcome, intention-to-treat as well

**Data Availability Statement:** According to Danish legislation, as long as the patient identification list (which combines the Civil Personal Registration (CPR) number with the patient-ID number) is still

existing, sharing of the data (in a de-identified format) is only allowed once permission from the Danish Data Protection Agency is obtained, which requires a signed data sharing agreement. When the permission for data processing expires (by May 16, 2028), deleting the patient identification list will fully anonymise data. Hereafter all collected data including the data dictionary are planned to be transferred to the Danish National Archives (https://www.sa.dk/en/), from where assess to the data can be granted to researchers who provide a methodologically sound proposal, and who are seeking to achieve aims described in the approved proposal. Proposals should be directed to mailbox@sa.dk Data requestors will need to sign a data sharing agreement. Additional material is available online in the form of the study protocol (https://www.ncbi.nlm.nih.gov/pubmed/26492879), and the informed consent form www.ctp-net.dk.

**Funding:** JC received 4 million DKK from TrygFonden for this study. Grant number: 102265 URL: https://www.trygfonden.dk/english Role of the funding source: The funders had no role in study design, data collection and analysis, decision to publish, or preparation of the manuscript.

**Competing interests:** The authors have declared that no competing interests exist.

as per-protocol analyses showed small but significant improvement on scores from pre- to post-treatment in all three groups but with no significant difference in improvement between groups.

## Conclusions

The findings do not provide evidence that either BBAT or mixed physical activity as add-on treatment bring significantly larger improvement on symptoms of PTSD compared to TAU alone for adult, trauma-affected refugees. There is a need for studies on potential subpopulations of trauma-affected refugees who could benefit from physical activity as a part of their treatment.

## Introduction

As the global refugee population is at its highest level ever recorded, with a refugee population of 22.5 million at the end of 2016 [1], the pertinence of issues regarding trauma-affected refugees is higher than ever. With an average prevalence of 30% of trauma-related psychiatric illnesses of PTSD and depression in populations from areas of conflicts [2] and the prevalence of comorbid pain as high as 80–100% among treatment-seeking trauma-affected refugees [3], the complex combination of PTSD, comorbid depression and chronic pain constitutes an often debilitating condition [4]. However, few rigorous studies exist on treatment effects for trauma-affected refugees. This is problematic because of the complex conditions for trauma-affected refugees, including prolonged trauma and post-migration stressors, entailing that generalising findings from non-refugee populations to trauma-affected refugees should be done with caution [4,5].

Based on the mutual maintenance theory [6], chronic pain in PTSD populations is a maintaining and sustaining factor in PTSD [6,7]. Consequently, the treatment and rehabilitation of patients with PTSD should include an integrated assessment and treatment of pain [7] as well as increasing activity levels [6–8]. In non-refugee PTSD populations, physical acitivty (PA) as intervention has gained growing attention in recent years [9,10]. While the only Cochrane review on PA as treatment for PTSD (in non-refugee populations) from 2010 concluded that no studies fulfilled the inclusion criteria [10], a more recent systematic review and meta-analysis from 2015 on four RCTs on PA for adult, non-refugee populations with PTSD found PA to be significantly more effective in decreasing PTSD and depressive symptoms compared to control conditions. This review, including both exercise and mind-body interventions, found promising results for both types of PA and suggests that PA may be a useful adjunct to standard treatment for people with PTSD [9]. To date, there is no evidence on the effect of physical activity (PA) as a part of treatment for trauma-affected refugees. Nonetheless, physical activity in various forms is widely used in addition to or as an integrated part of the treatment offered to this group [11,12]. There is thus an urgent need for examining the effectiveness of physical activity in treating PTSD and pain in robust, clinical RCTs.

Physical activity in this study, was defined as suggested by Caspersen et al. as "any bodily movement produced by skeletal muscles that results in energy expenditure" [13, page 126]. Furthermore PA can be divided in two categories: 1. structured exercise [13] and 2. mind-body interventions [14]. The two categories of PA have different content and methodology, and different theories of working mechanisms, although overlap exists [15]. Moreover, the focus on body awareness in mind-body interventions has obtained special attention in the PTSD literature due to theories of body awareness being a central aspect of the treatment of PTSD [16,17]. Due to the possible different working mechanisms between structured exercise

and mind-body interventions in the treatment of PTSD, in this study, we were interested both in studying if adding PA to treatment as usual (TAU) would increase treatment outcome but also if one of these two categories of PA was better than the other. As a mind-body intervention we were interested in studying basic body awareness therapy (BBAT), a mild, body-awareness-oriented physiotherapeutic method [18–20]. Improved body awareness has been suggested to be a potential mechanism for the therapeutic effect of mind-body therapies, such as BBAT [17,21] and BBAT is further one of the most widely used types of PA in the treatment of trauma-affected refugees in Denmark. BBAT has been tested among trauma-affected refugees in a pilot study and in a qualitative study and both found BBAT to add value to the treatment, e.g. improve symptoms of depression and anxiety [22] and improve sleep [23]. Nevertheless, evidence of the effect of BBAT is still lacking in trauma-affected refugees. We futher chose to study a PA based on structured exercise (in this study called mixed physical activity) as we were interested in studying two different types of PA from each end of the spectrum of PA. The aim of the present study was therefore to investigate if adding either BBAT or mixed physical activity to the treatment as usual (TAU) for trauma-affected refugees with PTSD would increase the treatment effect compared to TAU alone. It was hypothesised that adding physical activity to TAU would augment the treatment effect with respect to mental health symptoms (PTSD, depression, and anxiety), pain, quality of life as well as functional capacity, and body awareness.

## Methods

### Study design

The study was a 3-armed, pragmatic randomised controlled trial. It was an open-label, parallel group superiority study, allocation ratio of 1:1:1. The trial was conducted at the Competence Centre for Transcultural Psychiatry (CTP), a specialist outpatient clinic treating trauma-affected refugees in the Capital Region of Denmark. Since 2009, CTP has been conducting a series of randomised controlled pragmatic trials [24–26]. The target population at CTP comprises trauma-affected refugees (approximately 75%) and migrants with other psychiatric illnesses (approximately 25%). Only the former were included in the present study.

The study was registered with Clinicaltrials.gov (NCT01955538) October 7, 2013. The brief delay in registration with Clinicaltrials.gov (data inclusion starting September 13, 2013) was due to formalities in the registration process. The study was approved by the Ethics Committee of the Capital Region of Denmark July 16, 2013 (H-3-2013-080) as well as the Danish Data Protection Agency (02481 RHP-2013-024). The trial was monitored by Good Clinical Practice (GCP) Unit at Copenhagen University Hospital during the entire study period. The authors confirm that all ongoing and related trials for this intervention are registered. A paper on the study protocol is available: https://www.ncbi.nlm.nih.gov/pubmed/26492879 [27].

### Participants

All patients referred to CTP between September 13, 2013, and September 30, 2015 (except for a break in inclusion from October 1, 2014—December 31, 2014), were invited for a 1-3-hour pre-treatment interview with a medical doctor, recording the patient's history of psychiatric symptoms, trauma and social background, and evaluating for eligibility against the following criteria: 18 years or above; recognised as a refugee or family reunified with a refugee; diagnosed with PTSD according to ICD-10 research criteria; having experienced a psychological trauma in the past (e.g. imprisonment, torture, political persecution or war experiences); and motivated for treatment. Written informed consent was obtained from all participants.

The exclusion criteria were: having a psychotic disorder (defined as patients with an ICD-10 diagnosis F2x and F30.1-F31.9); a current abuse of drugs or alcohol (F1x.24-F1x.26) [28]; in

need of admission to a psychiatric hospital; physical handicaps that made participation in the physical activity impossible; a cardiac arrhythmia identified on the electrocardiogram taken before start of the treatment; or symptoms of heart problems that needed further examination.

Based on pre-treatment interview with a medical doctor, PTSD, depression and personality change after catastrophic event were diagnosed according to a diagnostic algorithm following ICD-10 criteria [28]. To exclude psychotic diagnoses, all patients were interviewed with relevant chapters from the Schedules for Clinical Assessment in Neuropsychiatry (SCAN interview) [29], by medical doctors who were certified SCAN raters.

All patients fulfilling the inclusion criteria and no exclusion criteria were invited to participate in the study. Patients who did not wish to participate in this study were offered treatment as described in the TAU manual. The complete date range from start of inclusion of participants to completion of treatment programme for all participants was September 13, 2013—January 24, 2017. The pragmatic design of this study primarily consisted of broad inclusion of the target group, only few exclusion criteria and a manualised flexible intervention offered at an outpatient clinic. All patients in need of an interpreter received this assistance and if possible, the same interpreter was used throughout the treatment.

## Randomisation and masking

Randomisation was conducted by sequentially, numbered, sealed envelopes, stratified by gender and level of PTSD symptoms (a score < or > 3.2 on the Harvard Trauma Questionnaire (HTQ)) by staff unconnected to patient treatment. The Department of Biostatistics at University of Copenhagen, not otherwise involved in the trial, produced a computer-generated randomisation sequence and drew up an anonymous randomisation list. Blinding the intervention was deemed impossible for clinicians and patients due to the nature of the intervention. However, Hamilton Depression and Anxiety (HAM D + A) interviews were conducted before and after treatment by a team of medical students, blinded to intervention group and time of the interview. These raters were trained at CTP and took part in joint ratings every 6–8 weeks to ensure high quality and interrater reliability. The clinicians were not given access to the results of the Hamilton interviews.

## Interventions

The three randomisation groups were:

1. TAU alone (control group = group C)

2. TAU + basic body awareness therapy (group B)

3. TAU + mixed physical activity (group M)

Both add-on treatments comprised individual physiotherapy sessions, one hour a week for 20 weeks. All patients in the two physical activity interventions were encouraged to do home exercises of the relevant physical activity. All clinicians followed profession-specific manuals developed for the target group.

**Treatment as usual (TAU).** The patients were all offered TAU. TAU consisted of approximately 6–7 months (planned) interdisciplinary treatment within a framework of 10 sessions with a medical doctor, and 16 sessions with a psychologist. TAU was divided into two phases. During the first phase, the patient had weekly sessions with a medical doctor, during the second phase monthly sessions with the medical doctor, and weekly sessions with a psychologist. According to needs, typically 1–2 counselling sessions on relevant social issues with a social worker were offered.

In the sessions with the medical doctor the main focus was on initiating pharmacological treatment if needed and providing psychoeducation on a wide range of topics such as explaining symptoms of PTSD and depression; advice on how to improve sleep. When pharmacological treatment was initiated, it was done by following the clinic's algorithm, based on the present knowledge on pharmacological treatment of trauma-affected refugees.

The sessions with the psychologists consisted of one-hour sessions of individual flexible cognitive behavioural therapy (CBT) with elements of acceptance and commitment therapy, stress management and mindfulness. The psychologist manual was developed by the psychologists at CTP using the results and experiences from the recent RCTs at CTP [24,26]. See also Nordbrandt et al., 2015 [27] for further description of TAU.

**The physical activity interventions.** The intervention groups B and M were offered 20 weekly sessions with a physiotherapist starting in phase 1. Both physiotherapeutic interventions were mild forms of physical activity but with certain flexibility to regulate the intensity according to the individual patient's capability. Both interventions followed manuals developed in cooperation with physiotherapists experienced in working with the target group. All patients were encouraged to do exercises used in the intervention as homework [27]. Each specific exercise was described in the manual, and for every session the actual exercises used and level of participation performed was noted in the patient record. Both physiotherapeutic interventions had a method sheet outlining the different topics/themes, which the physiotherapy could cover, and the topics/themes were connected to a number of specific exercises. Both manuals are available at request from the last author.

- *Basic body awareness therapy (BBAT) (group B)*: The exercises comprised slow, guided movements while standing, sitting and lying down, aiming at normalising and improving balance, muscle tension, free breathing and awareness. Only certified BBAT physiotherapists were teaching BBAT.

- *Mixed physical activity (group M)*: The intervention included basic exercises focusing on improving strength, endurance, balance and coordination, using simple tools such as water bottles, bicycle, resistance bands, grocery bags etc. The purpose of the tools was that the exercises could easily and inexpensively be practiced at the home of the participants, and could be continued after completion of the treatment programme.

## Outcomes

The primary outcome was severity of PTSD symptoms measured on the self-administered Harvard Trauma Questionnaire (HTQ), developed for research on trauma-affected refugees and validated in several populations [30,31]. Secondary outcomes were the self-administered rating scales Hopkins Symptom Check List (HSCL-25), assessing the severity of anxiety and depression symptoms [32], the somatisation scale of Symptom Checklist-90 (SCL-90) [33], quality of life evaluated on WHO-5 [34] and the Sheehan Disability Scale (SDS) [35] for level of functioning. Observer ratings included the Global Assessment of Functioning for symptoms and functioning (GAF-S and -F) [36], HoNOS assessing health and social functioning [37], and the Hamilton Depression and Anxiety scales (HAM-D and HAM-A) [38], which were conducted by blinded assessors. Pain was assessed by two self-administered rating scales: Visual Analogue Scale (VAS); and Brief Pain Inventory short form (BPI) [39,40]. BPI assesses pain intensity and pain interference and contains a diagram for shading pain location. The pain interference domain is recommended by international consensus (IMMPACT statement) in clinical trials evaluating physical functioning in chronic pain [41]. Body awareness was assessed on the self-

administered rating scale Multidimensional Assessment of Interoceptive Awareness (MAIA) [17]. Functional fitness was assessed by three performance-based physiotherapeutic measures conducted by the physiotherapist: Dynamic Gait Index (DGI) [42], Senior Fitness Test [43], and De Morton Mobility Index (DEMMI) [44]. This range of outcomes was used to enable comparison of our results to the previous studies at CTP and to evaluate on aspects specifically related to PA and pain.

All self-administered outcomes were available in the five main languages of the patients (Danish, Arabic, English, Bosnian/Serbo-Croatian and Farsi). There were a few illiterate (number unknown) participants and also a few participants where the questionnaires had not been translated to their language. These participants were assisted by an interpreter.

Patients were asked to complete most self-administered outcomes three times during the treatment course: at the pre-treatment interview, shortly before initiating phase two and post-treatment. HAM-D and -A were conducted pre- and post-treatment. All three intervention groups had a pre- and a post-treatment assessment with a physiotherapist, mapping physical difficulties and injuries, and conducting the functional fitness tests. Patients completed the self-administered questionnaires MAIA and BPI at these assessments. In this paper, only pre- and post-treatment measurements will be analysed, i.e two data points.

All medical doctors, psychologists and physiotherapists were asked to report to the investigator if becoming aware of any serious adverse events during the trial, whether related to the intervention or not. All serious adverse events discovered were registered and reported to The National Committee on Health Research Ethics.

Concomitant medicine was registered at the start of treatment.

A follow-up was carried out six months after ending the treatment programme. These results will be reported elsewhere.

## Statistical analysis

Power calculations were conducted with regard to the primary outcome HTQ. The study was initially planned to include approximately 250 patients as a conservative estimate of 200 participants eligible for intention-to-treat analysis with about 65 in each of the three groups would provide power of 81% to detect a group difference corresponding to a standardised difference of ½ SD (Cohen's d) and power to detect a difference of 1 SD of close to 100% (using a 5% level of significance and planning to use contrasts to compare group means in case of a significant overall test of group differences). A difference of less than ½ SD was considered less relevant from a clinical perspective. However, due to a smaller number than expected of male patients with a high HTQ score (which we stratified for in accordance with the average level of HTQ-score in previous RCTs conducted at CTP [24,25]) a larger number was included. Thus, the final number of included patients was 338.

Pre-treatment characteristics and descriptive data on the treatment were analysed for group differences by Chi-square test with Fisher's exact test and one-way ANOVA.

Pre- and post-treatment ratings were analysed in a mixed model including intervention group and rating time (pre-treatment vs post-treatment) as well as the interaction between time and intervention group. By using Stata's commands "margins" and "contrast", it was possible to estimate means of the treatment groups for pre- and post-treatment ratings and to test pre- post treatment differences in ratings within groups and between-groups differences in pre-post-treatment differences in ratings (corresponding to tests of the interaction between intervention group and rating time). This analysis was carried out both as the primary intention-to-treat analyses of all participants who completed pre-treatment ratings, and in addition on a reduced sample (per-protocol analyses). The per-protocol population was defined as all

patients in the control group plus all from B and M who participated in ≥10 physiotherapy sessions.

Two-tailed tests with a 5% level of significance were used in all statistical tests. Robust standard errors were used for conducting the mixed model analyses. Data was entered in the database via double data entry. All analyses were performed using STATA/SE 14.2 for Windows.

## Results

A total of 839 patients were screened for the trial. Of these, 338 fulfilled all inclusion criteria and no exclusion criteria and were included in the trial and randomised to either control group (C), basic body awareness therapy (B) or mixed physical activity (M) (C/B/M, n = 110/ 114/114). Of the 338, 20 patients were excluded during the study period due to exclusion criteria. Thus, 318 patients were eligible for intention-to-treat analyses (ITT-analyses) and 228 patients were eligible for per-protocol analysis (PP-analysis), see Consort Flow Chart, Fig 1. Mean length of the total treatment course was ten months. Respectively 23, 22 and 32 (C/B/M) patients dropped out of treatment before completing the post-treatment assessment. Average attendance of physiotherapy was ten sessions for both B and M. In the per protocol sample the average number of physiotherapy session in both groups B and M was 14.

Table 1 illustrates the baseline characteristics of the study population regarding demographics, trauma history, mental health and medication. In addition to PTSD, 294 (97%) of the patients had a comorbid depressive disorder. The functional fitness level on the Senior Fitness Test and DEMMI showed a mean corresponding to 80-90-year-olds at baseline on most subscales. The high levels of further comorbidity appear from Table 1.

Table 2 illustrates the mixed model analyses on the ITT-sample. On pre-treatment rating scores, there were no significant differences between the groups except for one outcome; a subdomain on the Brief Pain Inventory, the BPI interference (p = 0.0196) (Table 2).

On the primary outcome HTQ, we found no post-treatment group differences in scores, corresponding to the overall test of mean differences. However, we found significant decline in HTQ scores for all three intervention groups between pre- and post-treatment ratings. The decline was similar in all three groups. Accordingly, there were no significant group differences in change over time (p = 0.8573, Table 2).

For the secondary outcome measures in Table 2 (and S1 Table) the results were similar. There were no significant pre- or post-treatment group differences, and although most variables showed significant improvement in rating scores over time, there were no significant group differences in changes from pre- to post-treatment scores.

For the DEMMI measure only, the group differences were approaching significance (p = 0.0837) but this reflected deterioration in all three groups (S1 Table).

The findings of the mixed model PP-analyses (S2 Table) were consistent with the ITT-analysis, which showed no differences between groups on the primary outcome (HTQ); the same tendency of general improvement in all three groups; and no significant differences between groups, except on the BPI shaded score and DEMMI score.

During the study period there were no incidents of serious adverse events related to the interventions. Three patients (all from group B) withdrew from the physical activity and reported it was due to experiencing a deterioration of physical pain which they found related to engaging in the BBAT. Among those patients completing the full treatment program, 14 patients (C/B/M = 1/7/6) reported that their pre-existing physical pain had deteriorated after starting the treatment; expressing either a greater extent of discomfort or direct difficulties taking part in certain elements of the treatment.

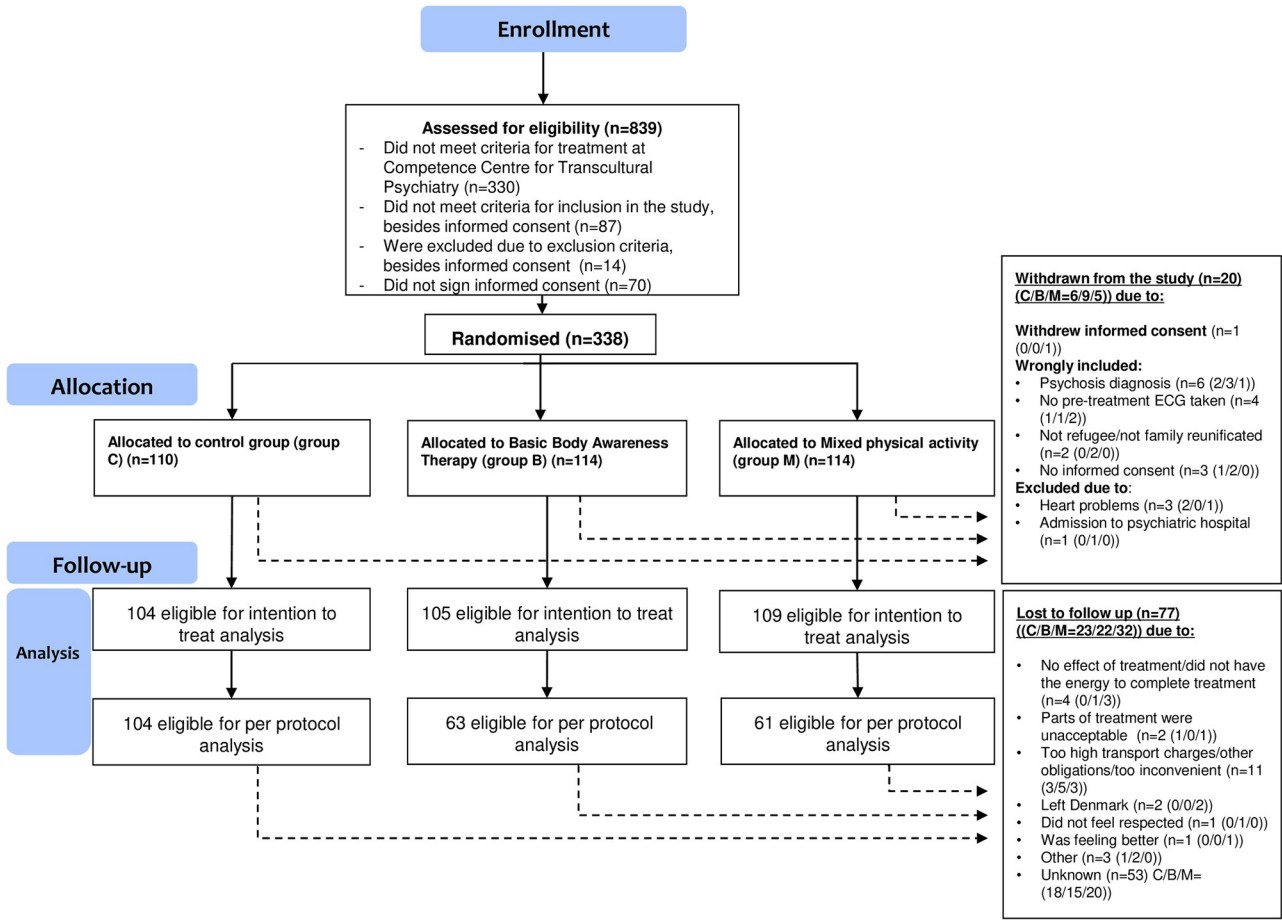

**Fig 1. Consort 2010 flow diagram PTF4.**

## Discussion

To our knowledge, this pragmatic RCT is by sample size the largest RCT yet conducted on trauma-affected refugees in an interdisciplinary, clinical setting. Furthermore, we believe it is the first RCT to compare two types of physical activity with TAU for trauma-affected refugees.

The study found overall small but significant improvement over time on all three groups on the primary and on four of the secondary outcomes (WHO, GAF-S, HoNOS, MAIA -not worrying) but in contrast to our hypothesis, we found neither BBAT nor mixed physical activity to significantly increase the treatment effect compared to TAU alone.

The results from the present study seem to differ from the conclusion in a review including four studies of PA as adjunctive treatment for non-refugee populations with PTSD [9], which points at an improved treatment effect when using PA as adjunctive treatment. The total number of sessions with PA in the studies included in the review do not differ substantially from the total number of sessions in our study. However, the four populations included in the review are different from the present population, as they are adult veterans; civilians; inpatients; and online recruited. Whether they differed in chronicity from the present population is not clear. However, as mentioned, due to the number of and type of traumas as well as post-migratory stressors that trauma-affected refuges face, conclusions based on other populations do not necessarily apply to trauma-affected refugees. Several studies have previously suggested

**Table 1. Pre-treatment characteristics of the control and intervention groups.**

| Pre-treatment characteristics | All (N = 318) | Control (N = 104) | Basic body awareness therapy (N = 105) | Mixed physical activity (N = 109) |
|---|---|---|---|---|
| | Mean (SD) | | | |
| **Demographic information** | | | | |
| **Age** | 44.6 (10.3) | 46.2 (10.4) | 43.1 (10.7) | 44.6 (9.5) |
| **Years since arrival in Denmark (n = 315)** * | 15.2 (8.6) | 15.7 (9.3) | 15.1 (8.0) | 14.9 (8.5) |
| | N (%) | | | |
| **Male gender** | 150 (47.2) | 47 (45.2) | 49 (46.7) | 54 (49.5) |
| **Female gender** | 168 (52.8) | 57 (54.8) | 56 (53.3) | 55 (50.5) |
| **Refugee camp before arrival in DK (n = 283)** * | 66 (23.3) | 23 (25.3) | 24 (24.7) | 19 (20.0) |
| **Country of origin** | | | | |
| Afghanistan | 53 (16.7) | 12 (11.5) | 20 (19.0) | 21 (19.3) |
| Chechnya | 11 (3.5) | 5 (4.8) | 3 (2.9) | 3 (2.8) |
| Former Yugoslavia | 23 (7.2) | 8 (7.7) | 8 (7.6) | 7 (6.4) |
| Iran | 32 (10.1) | 9 (8.7) | 9 (8.6) | 14 (12.8) |
| Iraq | 99 (31.1) | 33 (31.7) | 33 (31.4) | 33 (30.3) |
| Lebanon | 34 (10.7) | 14 (13.5) | 11 (10.5) | 9 (8.3) |
| Somalia | 11 (3.5) | 4 (3.8) | 1 (1.0) | 6 (5.5) |
| Syria | 25 (7.9) | 10 (9.6) | 7 (6.7) | 8 (7.3) |
| Other | 30 (9.4) | 9 (8.7) | 13 (12.4) | 8 (7.3) |
| **Trauma history** | | | | |
| Torture | 130 (40.9) | 44 (42.3) | 39 (37.1) | 47 (43.1) |
| Imprisonment | 140 (44.0) | 45 (43.3) | 42 (40.0) | 53 (48.6) |
| Soldier (n = 314) * | 83 (26.4) | 26 (25.5) | 29 (27.6) | 28 (26.2) |
| Sexual violence (n = 226) * | 33 (1.6) | 12 (17.1) | 11 (1.3) | 10 (12.7) |
| Violence from relatives (n = 255) * | 85 (33.3) | 28 (35.0) | 36 (40.9) | 21 (24.1) |
| Cranial trauma with loss of consciousness (n = 118) * | 97 (82.2) | 31 (76.6) | 33 (78.6) | 33 (94.3) |
| **Psychosocial status** | | | | |
| Presently employed/studying (n = 305) * | 26 (8.5) | 10 (9.9) | 9 (9.0) | 7 (6.7) |
| Living alone all the time (n = 307) * | 74 (24.1) | 25 (24.0) | 21 (20.8) | 28 (27.5) |
| Having children of < 18 years of age (n = 305) * | 175 (57.4) | 63 (61.8) | 55 (55.6) | 57 (54.8) |
| Duration of education from home country (years) (n = 289) * | | | | |
| <2 | 12 (4.1) | 5 (5.1) | 7 (7.5) | 0 (0.0) |
| 2–5 | 25 (8.6) | 9 (9.1) | 8 (8.6) | 8 (8.2) |
| >5–10 | 89 (30.7) | 31 (31.3) | 26 (28.0) | 32 (32.7) |
| >10–15 | 123 (43.4) | 36 (36.4) | 43 (46.2) | 44 (44.9) |
| >15 | 40 (36.8) | 18 (18.2) | 9 (9.7) | 13 (13.3) |
| Needing translator during medical doctor sessions (n = 295) * | 184 (62.4) | 58 (61.7) | 63 (62.4) | 63 (63.0) |
| **Diagnoses (ICD-10) additional to PTSD** | | | | |
| Enduring personality change after catastrophic experience (n = 182) * | 44 (24.2) | 17 (31.5) | 14 (20.9) | 13 (21.3) |
| Depression (n = 303) * | 294 (97.0) | 97 (98.0) | 100 (96.2) | 97 (97.0) |
| Other psychiatric disorder (n = 201) * | 18 (9.0) | 4 (6.8) | 6 (8.1) | 8 (11.8) |
| Psychiatric symptoms since ≥ 10 years (n = 292) * | 183 (62.7) | 57 (60.0) | 63 (63.0) | 63 (65.0) |
| **Concurrent medicine at beginning of treatment (categorised according to WHO ATC/DDD Index 2018):** | All (N = 318) | C (N = 104) | B (N = 105) | M (N = 109) |
| | N (%) | | | |
| **Antidepressants total (N06A)** | 102 (32.1) | 36 (34.6) | 32 (30.5) | 34 (31.2) |
| SSRI | 64 (20.1) | 23 (22.1) | 20 (19.1) | 21 (19.3) |
| SNRI | 19 (6.0) | 9 (8.7) | 5 (4.8) | 5 (4.6) |

*(Continued)*

**Table 1.** (*Continued*)

| Pre-treatment characteristics | All (N = 318) | Control (N = 104) | Basic body awareness therapy (N = 105) | Mixed physical activity (N = 109) |
|---|---|---|---|---|
| NaSSA | 31 (9.80) | 9 (8.7) | 8 (7.6) | 14 (12.8) |
| Tricyclic antidepressants | 14 (.4) | 5 (4.8) | 7 (6.7) | 2 (1.8) |
| **Antipsychotics (N05A)** | 28 (8.8) | 8 (7.7) | 14 (13.3) | 6 (5.5) |
| **Anxiolytics (N05B)** | 11 (3.5) | 3 (2.9) | 4 (3.8) | 4 (3.7) |
| **Hypnotics and sedatives (N05C)** | 24 (7.6) | 7 (6.7) | 10 (9.5) | 7 (6.4) |
| **Painkillers/analgesics total** | 140 (61.4) | 41 (39.4) | 46 (43.8) | 53 (48.6) |
| Opioids (N02A+ R05DA04) | 44 (13.8) | 16 (15.4) | 14 (13.3) | 14 (12.8) |
| Salicylic acids + other analgesics, antipyretics (NSAID, paracetamol) (N02B) | 109 (34.3) | 30 (28.9) | 37 (35.2) | 42 (38.5) |
| Triptanes (N02C) | 19 (6.0) | 6 (5.8) | 4 (3.8) | 9 (8.3) |
| Antiepileptics used as analgesics (N03A) | 12 (3.8) | 2 (1.9) | 7 (6.7) | 3 (2.8) |

*SD* standard deviation.

*Data not available for all randomised participants

that treatment effects in trauma-affected refugees are typically smaller than in other trauma-affected populations[24–26, 45].

## A question of chronicity?

The question remains whether the difference between the results of the above-mentioned studies and the present study can be explained by the level of chronicity, reflected in number of years in the receiving country at the time of treatment start. In a previous study in the same setting, Sonne et al.[46] found improvement in self-reported depression in trauma-affected refugees to be negatively correlated to long duration of mental problems. The mean number of years in Denmark (15.2 years (SD 8.60)) and nearly 63% (n = 183) having psychiatric symptoms for more than 10 years are therefore important facts to consider when comparing the present study to other treatment outcome studies with PTSD populations with less chronicity and comorbidity.

In the Cochrane review from 2017 on PA for chronic pain, the authors concluded that the favourable effects are inconsistent throughout the trials, and with mostly small to moderate effects, and that the present evidence is based on small studies of varying quality [47]. Hence, clinically based evidence on PA for pain is still insufficient and the results from this study contribute to filling this knowledge gap.

**Strengths and limitations.** The study has important strengths. The sample size of the study makes the results robust, and as the study was a pragmatic trial with broad inclusion criteria conducted in a clinical setting, results of the trial are likely to have a high generalisability to other clinical refugee health care settings. Comparing two different physical activity interventions to a TAU-only-group was a unique possibility to compare the two interventions internally as well as comparing them to TAU. Additionally, BBAT as method was prior to this study tested in a pilot study on the same target group, and the method was found satisfactory and acceptable [22]. In the pilot study, BBAT was offered in group sessions, which was the reason for several patients not wishing to participate in the study. In the present study we therefore avoided group sessions as this had been reported to be an obstacle for participation. By using individual treatment sessions, no potential spillover effect of a group could blur the

**Table 2. Score differences between pre-treatment ratings and post-treatment ratings on the Intention-to-treat population.**

| Rating | Groups and differences | Mean pre-treatment score (SE) | Mean post-treatment score (SE) | Difference (SE) | P-value |
|---|---|---|---|---|---|
| **HTQ** | C | 3.17 (0.04) | 2.96 (0.06) | **-0.21 (0.06)** | 0.0003** |
| | B | 3.17 (0.04) | 3.00 (0.07) | **-0.17 (0.06)** | 0.0024** |
| | M | 3.18 (0.04) | 2.97 (0.07) | **-0.21 (0.07)** | 0.0015** |
| | Difference, p-value | 0.9778 | 0.9116 | 0.8573 | - |
| **HSCL-25** | C | 3.02 (0.051) | 2.80 (0.07) | **-0.22 (0.07)** | 0.0017** |
| | B | 2.95 (0.05) | 2.84 (0.08) | **-0.11 (0.06)** | 0.0907 |
| | M | 2.98 (0.05) | 2.86 (0.08) | **-0.11 (0.08)** | 0.1455 |
| | Difference, p-value | 0.6246 | 0.8238 | 0.4546 | - |
| **SDS** | C | 22.61 (0.62) | 22.38 (0.84) | **-0.23 (0.86)** | 0.7886 |
| | B | 22.40 (0.63) | 21.80 (0.83) | **-0.60 (0.81)** | 0.4662 |
| | M | 23.66 (0.54) | 21.12 (0.89) | **-2.54 (0.86)** | 0.0032** |
| | Difference, p-value | 0.2536 | 0.5865 | 0.1211 | - |
| **WHO-5** | C | 14.93 (1.59) | 23.18 (2.56) | **8.25 (2.37)** | 0.0005** |
| | B | 16.82 (1.47) | 23.52 (2.52) | **6.70 (2.44)** | 0.0061** |
| | M | 16.19 (1.491) | 25.87 (2.845) | **9.68 (2.79)** | 0.0005** |
| | Difference, p-value | 0.6768 | 0.7520 | 0.7205 | - |
| **Ham-D** | C | 22.73 (0.65) | 21.75 (0.77) | **-0.98 (0.76)** | 0.1995 |
| | B | 22.58 (0.53) | 21.94 (0.83) | **-0.64 (0.70)** | 0.3606 |
| | M | 21.45 (0.72) | 20.42 (0.86) | **-1.03 (0.78)** | 0.1869 |
| | Difference, p-value | 0.3464 | 0.3765 | 0.9171 | - |
| **Ham-A** | C | 26.82 (0.85) | 26.79 (0.96) | **-0.03 (1.06)** | 0.9789 |
| | B | 27.03 (0.7) | 26.32 (1.0) | **-0.71 (0.94)** | 0.4500 |
| | M | 25.53 (0.94) | 25.87 (1.14) | *0.34 (0.98)* | 0.7278 |
| | Difference, p-value | 0.4180 | 0.8260 | 0.7342 | - |
| **GAF-F** | C | 51.63 (0.87) | 55.05 (1.22) | **3.42 (1.31)** | 0.0086** |
| | B | 52.47 (0.84) | 54.62 (1.10) | **2.15 (1.02)** | 0.0348* |
| | M | 50.94 (0.78) | 54.58 (1.46) | **3.64 (1.31)** | 0.05990 |
| | Difference, p-value | 0.4087 | 0.9564 | 0.5990 | - |
| **GAF-S** | C | 51.27 (0.68) | 55.89 (1.15) | **4.62 (1.20)** | 0.0001** |
| | B | 51.88 (0.64) | 54.64 (0.93) | **2.76 (0.86)** | 0.0014** |
| | M | 51.04 (0.69) | 55.41 (1.45) | **4.37 (1.34)** | 0.0011** |
| | Difference, p-value | 0.6493 | 0.6889 | 0.3667 | - |
| **BPI severity** | C | 6.37 (0.18) | 6.58 (0.21) | *0.21 (0.19)* | 0.2853 |
| | B | 6.16 (0.20) | 6.39 (0.22) | *0.23 (0.18)* | 0.1912 |
| | M | 6.49 (0.20) | 6.71 (0.26) | *0.22 (0.20)* | 0.2569 |
| | Difference, p-value | 0.5112 | 0.6144 | 0.9965 | - |
| **BPI interference** | C | 7.81 (0.17) | 7.54 (0.21) | **-0.27 (0.23)** | 0.2281 |
| | B | 7.06 (0.24) | 6.95 (0.28) | **-0.11 (0.24)** | 0.6407 |
| | M | 7.24 (0.25) | 7.16 (0.28) | **-0.07 (0.23)** | 0.7463 |
| | Difference, p-value | 0.0196* | 0.2250 | 0.7992 | - |

SE = standard error

* p ≤.0.05

** p≤.0.01

**Bold** = Improvement, *Italic* = Deterioration

HTQ, HSCL-25 = 1–4 (1 best score), SDS = 0–10 (0 best score), WHO-5 = 0–100 (100 best score), HAM-D = 0–52 (0 best score), HAM-A = 0–56 (0 best score), GAF-S/-F = 0–100 (100 best score), BPI severity/interference = 0–10 (0 best score).

*HTQ* Harvard Trauma Questionnaire, *HSCL-25* Hopkins Symptom Checklist-25, *SDS* Sheehan Disability Scale, *WHO-5* WHO-5 Well Being Index, *HAM-D/-A* Hamilton Depression/Anxiety Rating scales, *GAF-F/-S* Global assessment of Functioning (Symptom/Function), *BPI severity/interference* Brief Pain Inventory severity/interference.

All means and SEs in Table 2 are mixed model estimates and so are the p-values. The p-values in the pre- and post-treatment columns refer to an overall test of the significance mean differences between the three groups (corresponding to an ANOVA F-test). The p-values in the column showing pre-post treatment differences refer to an overall test of the significance of group differences in mean pre-post treatment differences (this test of group differences in pre-post treatment difference corresponds to the statistical test of the time by intervention interaction). Finally, the column with p-values shows p-values corresponding to the pre-post treatment mean difference in each of the three groups.

results. Furthermore, the number of sessions in the trial was comparable to other PTSD PA intervention studies [9].

However, there were also certain limitations to the study. The attendance rate to the offered PA sessions was rather low. Further, the physical activity in the present study was mild, individually adapted, and with a frequency of once a week (with a maximum of 20 weekly sessions), as it was tailored to fit the functional capacity of the patients. As the level of exercise intensity reached in this study therefore was relatively low and no significant improvements on the functional fitness outcomes were seen, one might ask, if this could explain the lack of effect of M. However, regarding effectiveness, systematic reviews of PA for depression and PTSD [9,48] show conflicting evidence of effects of high-moderate intensity vs. light-moderate physical activity. As this study was pragmatic, the results do not reflect what is possible in a sample of non-chronic PTSD patients with no comorbidity but contribute to our knowledge about what may realistically be achieved in target groups of chronically ill, trauma-affected refugees. Results from previous randomised studies, the pilot BBAT study as well as clinical experience with the target group, had implications for the design of the present study. The frequency and intensity of the PA was adapted to avoid too many dropouts from treatment. Due to a number of patient-related reasons, including costs for transport to CTP, which was not reimbursed, we could not plan for more than one session a week. Apart from frequency of sessions, also the intenstity of the PA in the sessions was adapted to the capacity of the target group. Manuals were developed based on knowledge and clinical experience with the target group and although a more frequent and intense PA could have been wished for this was not possible. With a higher intensity of PA in the study it is expected that a larger number of the patients would not have been able to participate, while a few might have had more benefit. Future studies that could overcome some of these challenges could look at interventions designed to enhance motivation and thereby attendance to physical activity. Studies on motivational interventions have shown promising results in patients with schizophrenia [49].

That said, it does not seem that the frequency alone accounts for the lack of difference between the physiotherapy groups and the TAU group in our study since the PP analyses did not show any difference in between groups either, despite having a higher attendance to physiotherapy sessions (14 sessions in ITT-sample versus ten sessions in PP-sample). Hence, future studies may rather explore potential subpopulations who could benefit from physical activity as a part of their treatment as the present study points to no add-on effect for the clinical populations of trauma-affected refugees with a high degree of comorbidities and chronicity.

The questionnaire MAIA was not pre-tested on the target group in a pilot study, and since some patients reported difficulties understanding the MAIA questions, it raises the issue of the validity of the MAIA results. Also, the diagnosis of PTSD was made based on clinical interviews instead of a validated, structured interview. Furthermore, it was not deemed possible to blind patients or clinicians. Important to note is also that B and M were receiving a much more extensive treatment course than the control group (C), making C less comparable to group B and M regarding the amount of time with a clinician. Meanwhile, bearing in mind the difference in time with a clinician and despite this a lack of differences in treatment effect between the control group versus group B and M, the results of the study become even more convincing.

The results of this present study are on a par with the results of the succession of randomised trials conducted at CTP [24–26], showing no superiority in decreasing symptoms of PTSD and about the same size in pre- to post-treatment differences. The previous studies all have similar severity of PTSD at baseline, very high depression comorbidity (>90%) as well as a similar length of time in the new country. While recognising that the present interventions were only two out of a range of existing PA interventions, this could indicate that on average,

in a target group with chronic conditions such as the present, only a certain treatment effect is possible, perhaps regardless of the content of the add-on treatment.

## Impact on future treatment

The lack of further improvement found in this study, when adding physical activity to TAU, should be taken into account when planning treatment and developing treatment guidelines for trauma-affected refugees with high levels of symptom severity, comorbidities and chronicity. With the rather low attendance found in this study, there is a need for more knowledge on how to motivate the patients for active participation in PA in order to further study if a higher intensity of treatment could increase the effectiveness. However, these efforts should take the symptom severity, comorbiditites and chonicity in this population into account. It was not deemed possible in this study to increase the intensity and still have a broad inclusion of participants. Thus, further research is needed on the target group, including on the relationship between pain and symptoms of PTSD and between chronicity and symptom improvement, to understand how we in the future can provide more effective treatment for trauma-affected refugees with a high degree of comorbidity and symptom chronicity.

## Supporting information

**S1 Table. Score differences between pre-treatment ratings and post-treatment ratings in the intention-to-treat population.**
(DOCX)

**S2 Table. Score differences between pre-treatment ratings and post-treatment ratings in the per-protocol population.**
(DOCX)

**S1 Checklist.**
(DOC)

**S1 Protocol.**
(DOC)

## Acknowledgments

The authors thank the senior consultant psychiatrist and head of CTP, Morten Ekstrøm, for his valuable support of the project. Also, a warm thank to all participants in the study and to all clinicians, secretaries, and data management team at CTP for their work during the study.

## Author Contributions

**Conceptualization:** Maja Sticker Nordbrandt, Charlotte Sonne, Erik Lykke Mortensen, Jessica Carlsson.

**Data curation:** Maja Sticker Nordbrandt, Charlotte Sonne, Erik Lykke Mortensen, Jessica Carlsson.

**Formal analysis:** Maja Sticker Nordbrandt, Charlotte Sonne, Erik Lykke Mortensen, Jessica Carlsson.

**Funding acquisition:** Maja Sticker Nordbrandt, Jessica Carlsson.

**Investigation:** Maja Sticker Nordbrandt, Erik Lykke Mortensen, Jessica Carlsson.

**Methodology:** Maja Sticker Nordbrandt, Charlotte Sonne, Erik Lykke Mortensen, Jessica Carlsson.

**Project administration:** Maja Sticker Nordbrandt, Jessica Carlsson.

**Resources:** Jessica Carlsson.

**Supervision:** Erik Lykke Mortensen, Jessica Carlsson.

**Validation:** Jessica Carlsson.

**Writing – original draft:** Maja Sticker Nordbrandt, Jessica Carlsson.

**Writing – review & editing:** Maja Sticker Nordbrandt, Charlotte Sonne, Erik Lykke Mortensen, Jessica Carlsson.

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
