## [Decision Letter · Decision Letter 0]

6 Aug 2019

PONE-D-19-16080

Trauma-affected refugees treated with basic body awareness therapy or mixed physical activity as augmentation to treatment as usual – a pragmatic randomised controlled trial

PLOS ONE

Dear Dr Carlsson,

Thank you for submitting your manuscript to PLOS ONE. After careful consideration, we feel that it has merit but does not fully meet PLOS ONE’s publication criteria as it currently stands. Therefore, we invite you to submit a revised version of the manuscript that addresses the points raised during the review process.

In particular the stitica aspects need extended revision following the comments of reviewer 3. Please also consider carefully all other points raised by the other reviewers. 

We would appreciate receiving your revised manuscript by September 10th. To enhance the reproducibility of your results, we recommend that if applicable you deposit your laboratory protocols in protocols.io, where a protocol can be assigned its own identifier (DOI) such that it can be cited independently in the future. For instructions see: http://journals.plos.org/plosone/s/submission-guidelines#loc-laboratory-protocols

We look forward to receiving your revised manuscript.

Kind regards,

Andrea Martinuzzi

Academic Editor

PLOS ONE

2. Thank you for submitting your clinical trial to PLOS ONE and for providing the name of the registry and the registration number. The information in the registry entry suggests that your trial was registered after patient recruitment began. PLOS ONE strongly encourages authors to register all trials before recruiting the first participant in a study.

a) your reasons for your delay in registering this study (after enrolment of participants started);

b) confirmation that all related trials are registered by stating: “The authors confirm that all ongoing and related trials for this drug/intervention are registered”.

Please also ensure you report the date at which the ethics committee approved the study as well as the complete date range for patient recruitment and follow-up in the Methods section of your manuscript.

Reviewers' comments:

Reviewer's Responses to Questions

**Comments to the Author**

1. Is the manuscript technically sound, and do the data support the conclusions?

Reviewer #1: Yes

Reviewer #2: Partly

Reviewer #3: Partly

2. Has the statistical analysis been performed appropriately and rigorously? 

Reviewer #1: Yes

Reviewer #2: Yes

Reviewer #3: No

3. Have the authors made all data underlying the findings in their manuscript fully available?

Reviewer #1: Yes

Reviewer #2: Yes

Reviewer #3: Yes

4. Is the manuscript presented in an intelligible fashion and written in standard English?

Reviewer #1: Yes

Reviewer #2: Yes

Reviewer #3: Yes

5. Review Comments to the Author

Reviewer #1: The research is carried out with a pragmatic approach. It is well designed and written. Even if the research did not confirm the initial hypothesis, I can remark the following consideration:

Description of the selected population:

The sample size of the research population is meaningful and well selected. The issue of immigration from a war affected areas is an important and discussed problem. That’s why this research could contribute to understand the nature of the needs of each individual immigrant. There is a great necessity of planning and implementing health and social programs for all those who are arriving in Europe during these recent years.

Descriptions of the therapeutic programs

While the clinical characteristics of the selected populations have been well described and assessed, before and after the therapeutic programs, the readers have not idea of the quality and the quantity of the activities carried out by the patients in addition to the therapy as usual (TAU). Either the physical activities (PA) and the basic body awareness therapy (BBAT) should be better described in order to design a pragmatic protocol for all those will try to improve and implement the intervention on these kind of patients.

Measuring the efficacy of the therapeutic programs

The research has selected a large number of rating scales. All of them have measured the efficacy of the therapeutic programs. Some were measuring the severity of PTSD and other symptoms of depression and anxiety; others the functional impairment and the pain. Some raining scale has been adopted in order to measure the mobility of patients with many physical limitations. I think the use of the International Classification of Functioning, Disability and Health (ICF WHO 2001) could be recommended to have a complete and satisfactory assessment.

Further investigations

The research found overall small but significant improvement over time on all three groups on the primary and on many secondary outcomes but at contrary to the hypothesis, the researchers didn’t find any difference between the control group (with only the TAU) and the experiment ones. Since the study was build up with a strong and well prepared study method, I am suggesting to repeat the research replacing the PA and the BBAT with some kind of participation activities within the local community. Even if appears that the researchers don’t care about the participation of the refugees to a local community I think this must be included as part of a therapeutic approach for people affected by PTSD and related outcomes.

Reviewer #2: First, I would like to compliment the authors with their study, which has been conducted thoroughly. The subject of the study is extremely relevant since it evaluates treatment for a target group that suffers tremendously and does not receive a lot of attention. It is of the utmost importance that the authors state that results from other groups suffering from PTSD are not always applicable to refugees with trauma-related disorders.

Furthermore, the addition of body- and movement-related interventions is of importance when dealing with trauma-related disorders, in which the body as locus of control is so severely damaged.

The manuscript is highly readable.

My questions and remarks are the following:

General comments

Given the fact that two of the intervention studied pertain physical activity, it seems important to define physical activity (see also Rosenbaum et al.). Of the four included trials in the Rosenbaum review, two used a yoga-based intervention.

It is unclear why two different physical activity interventions are offered. It is stated that it is important to evaluate differences between the two, but why are two different activities offered in the first place?

Rosenbaum et al. define physical activity as any bodily movement produced by skeletal muscle requiring energy expenditure, thereby citing Caspersen et al. (1985).

However, BBAT could also be regarded as a body- and movement psychotherapeutic intervention, not aiming in the first place at a better physical activity but on ameliorating body-awareness as first aim. See e.g. Erikson et al. (2007) : Body awareness therapy represents a body-oriented physiotherapeutic approach focusing on both the physical and psychological aspects of body dysfunctions; see also the description in Catalan-Matamoros et al. (2010) on BBAT.

In the title of the manuscript it says: basic body awareness therapy or mixed physical activity, giving the impression that these are differently aimed interventions. However, further in the manuscript they are both shared under physical activity.

Abstract

Line 22: It would be better to define the aim somewhat broader, thereby differentiating the two forms of physical activity and making clear already that there are 3 arms.

Line 28/29: these lines should be rewritten, making clear that both add-on treatments are individual physiotherapy treatments.

Conclusions: could some lines as to the why of the lack of difference be added? (discussion).

It think this would better reflect the importance of the study, although results do not show differences to TAU.

Introduction:

I would suggest to skip the reference to the retracted article (lines 60-65).

I would suggest to make a difference between BBAT and mixed physical activity.

The text now mentions BBAT as a body-awareness oriented therapy in line 67, but in the lines following the effect of physical activity is the main issue.

Then the aim of the study is stated as evaluating if either BBAT or mixed physical activity would increase the treatment effect. But I would think there are thoughts about the differences of the effect and it would strengthen the article to articulate some hypotheses regarding the effect of targeting body awareness in trauma.

Participants

It is not clear to me whether the patients are all outpatients, or may be in day-treatment or in-patients

Line 117: patients who did not wish to participate in this study were offered TAU. Does this mean they were not part of the 110 participants in TAU?? Otherwise it should be made clear in the following section about randomization that these patients (and how many) take part.

Interventions:

These remarks are connected to my other remarks about the difference between BBAT, especially its specific therapeutic goals, and mixed physical activity.

Therefore, I would recommend to describe the two arms separately and not under the common denominator of physical activity. By this subdivision the content of the manuscript will be more according to what is suggested in the title of the manuscript.

Results

Lines 244/245

It seems rather obvious that, given a total time of ten months and an average attendance of ten sessions, so physiotherapy once a month, no effect is to be expected. I think this point should get more attention in the discussion. A heightened frequency and more focus on attendance (attendance being a general problem, see also the literature on physical activity for people with schizophrenia for example) could be a recommendation for the future.

How many sessions were there for the intervention groups? I read 20 weekly sessions starting in phase one (line 159). Does this mean a total of 20 sessions?

Line 250: was the percentage of comorbid depression established by the medical doctor according to ICD 10 criteria compared to the Hamilton depression scores? This seems sensible since the comorbid depression is surprisingly high (even given the fact that depression is the main comorbid disorder in PTSD).

Table 2: The fitness results and the MAIA results are absent in this table, while they are crucial for the measurement of the add-on effects. So, it is not clear which measures are for what reason in the Supporting information.

Discussion

Lines 315-316: I would suggest to delete the remark about the interventions in this study not being different from those in the Rosenbaum review. Interventions as well as populations are all very different in the Rosenbaum review, and are also different from the ones in the present study.

345: duration is long, but frequency is a weakness!

348: frequency of once a week would suggest 10 months = possibly 40 sessions!

356-358 I would suggest to skip these lines and possibly look for other references. The authors rely too much on Rosenbaum et al.

Lines 368-371: the add-ons are different because they use body- and movement interventions and it is strange to suggest that the bodily aspects were already met in the TAU.

Lines 374 and further: I have the impression the authors are disappointed, which I can imagine, but I think it would be worthwhile to think about other options, such as frequency of the add-on treatment, motivational issues, etc.

Impact on future treatment.

The first lines should really be deleted. Adding physical activity 10 times in a ten month treatment and having no effect, does not justify this conclusion.

Reviewer #3: Present article describes a three-arm RCT to find the efficacy of physical activity as an added component in reducing symptoms of PTSD. Trial recruited 338 patients though there was a significant dropout in two arms. The description of trial is very clear with self-administered HTQ as the primary outcome. The study fall shorts in many other statistical and trial consideration, which I describe bellow.

1. Why the trial is “pragmatic” is not clear. Please describe it why or take the word out.

2. How many data points per subject is collected on the course of trial is not clear.

3. The reason for conducting three-arm superiority trial is not very clear. Note, inclusion of more number of arms increases the problem related to multiple testing. This has effect on sample size and type-1 error which I will describe next.

4. The power analysis described in line 211-221 is insufficient. Please report standardized effects size (Cohen’s d) as it is not clear what does mean by 0.5 SD or 1SD. If I interpret 0.5SD and as Cohen’s d=0.5 then it is a moderately large effect size, but in the acceptable range. 1.0 is too large an effect size. It should also describe what statistical test is used and using what statistical model. It is not clear if a main effect is powered or an interaction term!! Since more than two arm is present there is a possibility of two or more co-primary hypothesis, which will require adjustment for multiplicity as otherwise type-1 error will be inflated (>0.05). This may render sample projected to be inferior. Primary analysis model and the model used for sample size should be same or closely following each other.

5. Also please mention which software or simulation method is used to do the power analysis.

6. In line 225-228 many analysis methods are described, but I only see the result of pre-post type comparisons in the table 1 and table 2. Where is time by intervention result reported? Also line 234 is confusing as the result following the line is not reported. There is no table related to mixed-model ANOVA. If the study includes more than 3 observations per subject the mixed-effect model based ANOVA should be the primary result for determining superiority.

7. ITT and per-protocol analysis both are done and it is not clear which result should be more important. It is sometime confusing and not clear reasoning is mentioned for both.

8. A major concern near 50% dropout in Arm B and M as presented in Figure 1. From the reason for dropout it seems the dropout are non-ignorable as a result if proper care is not taken the results are prone to selection bias. Neither ITT not PPA can take care of this high attrition. A proper approach will be modelling the Missing Not at Random (MNAR) mechanism. I am really doubtful if this aspect is taken care of.

6. PLOS authors have the option to publish the peer review history of their article (what does this mean?). If published, this will include your full peer review and any attached files.

Reviewer #1: Yes: Marco Sala, MPhil MSc Rehabilitation Sciences

IRCCS Medea La Nostra Famiglia, Como Italy

Reviewer #2: Yes: Mia Scheffers

Reviewer #3: No

---

## [Author Response · Author response to Decision Letter 0]

6 Oct 2019

PONE-D-19-16080

Trauma-affected refugees treated with basic body awareness therapy or mixed physical activity as augmentation to treatment as usual – a pragmatic randomised controlled trial

PLOS ONE

Dear Editor and reviewers

Thank you for offering us to revise our manuscript based on the constructive reviewer comments.

Response to Journal requirements:

2. Thank you for submitting your clinical trial to PLOS ONE and for providing the name of the registry and the registration number. The information in the registry entry suggests that your trial was registered after patient recruitment began. PLOS ONE strongly encourages authors to register all trials before recruiting the first participant in a study.

a) your reasons for your delay in registering this study (after enrolment of participants started);

b) confirmation that all related trials are registered by stating: “The authors confirm that all ongoing and related trials for this drug/intervention are registered”.

Please also ensure you report the date at which the ethics committee approved the study as well as the complete date range for patient recruitment and follow-up in the Methods section of your manuscript.

Response to Journal requirements:

Thank you for pointing this out. We have now added:

• A brief explanation of the delay in registration at Clinical Trials in the Methods section. “The study was registered with Clinicaltrials.gov (NCT01955538) October 7, 2013. The brief delay in registration with Clinicaltrials.gov (data inclusion starting September 13, 2013) was due to formalities in the registration process.”

• In the same section we have added the sentence: “The authors confirm that all ongoing and related trials for this intervention are registered”.

• In the Methods section we have also specified the date for the ethics committee approval (July 16, 2013) as well as the complete date range for patient recruitment and follow-up (last follow-up being January 24, 2017).

Reviewer #1

The research is carried out with a pragmatic approach. It is well designed and written. Even if the research did not confirm the initial hypothesis, I can remark the following consideration:

Description of the selected population:

The sample size of the research population is meaningful and well selected. The issue of immigration from a war affected areas is an important and discussed problem. That’s why this research could contribute to understand the nature of the needs of each individual immigrant. There is a great necessity of planning and implementing health and social programs for all those who are arriving in Europe during these recent years.

Descriptions of the therapeutic programs

1.1: While the clinical characteristics of the selected populations have been well described and assessed, before and after the therapeutic programs, the readers have not idea of the quality and the quantity of the activities carried out by the patients in addition to the therapy as usual (TAU). Either the physical activities (PA) and the basic body awareness therapy (BBAT) should be better described in order to design a pragmatic protocol for all those will try to improve and implement the intervention on these kind of patients.

#1.1 Response: It is true that the description of the exact content of the manuals is not available. We have now added more information regarding the activities carried out and we have also added the following: “Both physiotherapeutic interventions had a method sheet outlining the different topics/themes, which the physiotherapy could cover, and the topics/themes were connected to a number of specific exercises. Both manuals are available at request from the last author.”

1.2: Measuring the efficacy of the therapeutic programs

The research has selected a large number of rating scales. All of them have measured the efficacy of the therapeutic programs. Some were measuring the severity of PTSD and other symptoms of depression and anxiety; others the functional impairment and the pain. Some raining scale has been adopted in order to measure the mobility of patients with many physical limitations. I think the use of the International Classification of Functioning, Disability and Health (ICF WHO 2001) could be recommended to have a complete and satisfactory assessment.

#1.2 Response: We do agree that the ICF contains important information in a population as the present. However, we do believe that most of the aspects of the ICF were covered with the extensive battery of measurements used and that we did succeed with getting a rather complete assessment of the participants. In subsequent studies to the present, we have included the WHODAS. The WHODAS has been described to be an operationalization of the ICF and easier to administer than the ICF.

1.3: Further investigations

The research found overall small but significant improvement over time on all three groups on the primary and on many secondary outcomes but at contrary to the hypothesis, the researchers didn’t find any difference between the control group (with only the TAU) and the experiment ones. Since the study was build up with a strong and well prepared study method, I am suggesting to repeat the research replacing the PA and the BBAT with some kind of participation activities within the local community. Even if appears that the researchers don’t care about the participation of the refugees to a local community I think this must be included as part of a therapeutic approach for people affected by PTSD and related outcomes.

#1.3 Response: We fully agree that the life outside a treatment facility is of large importance to this population. However, assessing community activities was outside the scope of this research study. We are presently preparing a randomized study where the intervention focuses on a close collaboration with the municipality, using the WHODAS and where a qualitative sub study will look at participation in the local community.

Reviewer #2

First, I would like to compliment the authors with their study, which has been conducted thoroughly. The subject of the study is extremely relevant since it evaluates treatment for a target group that suffers tremendously and does not receive a lot of attention. 

2.1: It is of the utmost importance that the authors state that results from other groups suffering from PTSD are not always applicable to refugees with trauma-related disorders.

#2.1 Response: We very much agree the existing evidence on treatment outcome suggests that results from other trauma-affected populations cannot always be transferred to trauma-affected refugees. This is pointed out in the Introduction in the sentence “However, few rigorous studies exist on treatment effects for trauma-affected refugees. This is problematic because of the complex conditions for trauma-affected refugees, including prolonged trauma and postmigration stressors, entailing that generalising findings from non-refugee populations to trauma-affected refugees should be done with caution.”. We have added reference to a paper (“From Pioneers to Scientists Challenges in Establishing Evidence-Gathering Models in Torture and Trauma Mental Health Services for Refugees” Carlsson et al, 2014), that discusses this issue in depth. 

Further a sentence has been revised and another added including a reference stressing the same issue in the Discussion: “However, as mentioned, due to the number of and type of traumas as well as post-migratory stressors that trauma-affected refuges face, conclusions based on other populations do not necessarily apply to trauma-affected refugees. Several studies have previously suggested that treatment effects in trauma-affected refugees are typically smaller than in other trauma-affected population.”

Furthermore, the addition of body- and movement-related interventions is of importance when dealing with trauma-related disorders, in which the body as locus of control is so severely damaged.

The manuscript is highly readable.

My questions and remarks are the following:

General comments

2.2: Given the fact that two of the intervention studied pertain physical activity, it seems important to define physical activity (see also Rosenbaum et al.). Of the four included trials in the Rosenbaum review, two used a yoga-based intervention. It is unclear why two different physical activity interventions are offered. It is stated that it is important to evaluate differences between the two, but why are two different activities offered in the first place?

#2.2 Response: This is an important issue to address. We have now added a definition of physical activity (PA) and have defined structured exercise and mind body interventions as two sub categories of PA used in the study. The definition of PA is the same as the one below by Caspersen et al.. We have included the definitions in the Introduction section which has been extensively revised. We also now explain in more detail in the Introduction why we chose to include both categories of physical activities in the current trial.

2.3: Rosenbaum et al. define physical activity as any bodily movement produced by skeletal muscle requiring energy expenditure, thereby citing Caspersen et al. (1985).

However, BBAT could also be regarded as a body- and movement psychotherapeutic intervention, not aiming in the first place at a better physical activity but on ameliorating body-awareness as first aim. See e.g. Erikson et al. (2007) : Body awareness therapy represents a body-oriented physiotherapeutic approach focusing on both the physical and psychological aspects of body dysfunctions; see also the description in Catalan-Matamoros et al. (2010) on BBAT.

In the title of the manuscript it says: basic body awareness therapy or mixed physical activity, giving the impression that these are differently aimed interventions. However, further in the manuscript they are both shared under physical activity.

#2.3 Response: We agree that the definitions could be more clearly described in the manuscript. We use “physical activity” as a concept covering all types of exercise (using the broad definition by Caspersen et al.) and “mixed physical activity” for the intervention developed covering the category: structured exercises. We have now tried to make this distinction clearer throughout the manuscript. Please see also #2.2 above.

Abstract

2.4: Line 22: It would be better to define the aim somewhat broader, thereby differentiating the two forms of physical activity and making clear already that there are 3 arms.

#2.4 Response: The aim in the abstract has been reframed as suggested and is now identical to the one in the paper: “The aim of the present study was to investigate if adding either BBAT or mixed physical activity to the treatment as usual (TAU) for trauma-affected refugees with PTSD would increase the treatment effect compared to TAU alone.”

2.5: Line 28/29: these lines should be rewritten, making clear that both add-on treatments are individual physiotherapy treatments.

#2.5 Response: These lines have now been rewritten in order to underline that there are two different individual physiotherapy treatments: “Participants were randomised to receive either individual physiotherapy (basic body awareness therapy (group B) or individual mixed physical activity (group M)) one hour/week for 20 weeks plus TAU, or TAU only (group C).”

2.6: Conclusions: could some lines as to the why of the lack of difference be added? (discussion).

It think this would better reflect the importance of the study, although results do not show differences to TAU.

#2.6 Response: We have added the following sentence, now also included in the Discussion in the Conclusion: “A large number of the participants in the study have a chronic mental condition, often difficult to treat There is a need for studies on potential subpopulations of trauma-affected refugees who could benefit from physical activity as a part of their treatment.”

Introduction:

2.7: I would suggest to skip the reference to the retracted article (lines 60-65).

#2.7 Response: We have now deleted the sentences regarding the retracted article as well as the reference.

2.8: I would suggest to make a difference between BBAT and mixed physical activity.

The text now mentions BBAT as a body-awareness oriented therapy in line 67, but in the lines following the effect of physical activity is the main issue.

#2.8 Response: This comment is partly addressed previously by clarifying the definition of physical activity (PA), BBAT and mixed physical activity and the text added in relation to comment 2.3. We have additionally added relevant references. Please see response #2.3.

Furthermore, we have tried to make the difference between the two PA clearer by revising and adding text in the last part of the Introduction: 

“The two categories of PA have different content and methodology, and different theories of working mechanisms, although overlap exists. Moreover, the focus on body awareness in mind-body interventions has obtained special attention in the PTSD literature due to theories of body awareness being a central aspect of the treatment of PTSD. Due to the possible different working mechanisms between structured exercise and mind-body interventions in the treatment of PTSD, in this study, we were interested both in studying if adding PA to treatment as usual (TAU) would increase treatment outcome but also if one of these two categories of PA was better than the other.

2.9: Then the aim of the study is stated as evaluating if either BBAT or mixed physical activity would increase the treatment effect. But I would think there are thoughts about the differences of the effect and it would strengthen the article to articulate some hypotheses regarding the effect of targeting body awareness in trauma.

#2.9 Response: The specific reasons for targeting body awareness are now further explained, partially by the revision described above in #2.8 and partially by further revising the last part of the Introduction including relevant references:”Improved body awareness has been suggested to be a potential mechanism for the therapeutic effect of mind-body therapies such as BBAT and BBAT is further one of the most widely used types of PA in the treatment of trauma-affected refugees in Denmark. a mild, body-awareness-oriented.”

Participants

2.10: It is not clear to me whether the patients are all outpatients, or may be in day-treatment or in-patients

#2.10 Response: All patients are out-patients. This is stated in Methods; Study design: “The trial was conducted at the Competence Centre for Transcultural Psychiatry (CTP), a specialist outpatient clinic treating trauma-affected refugees in the Capital Region of Denmark.“

2.11: Line 117: patients who did not wish to participate in this study were offered TAU. Does this mean they were not part of the 110 participants in TAU?? Otherwise it should be made clear in the following section about randomization that these patients (and how many) take part.

#2.11 Response: We are sorry about this confusion. We did not collect any longitudinal data on those that were not a part of the study. These patients were offered the same treatment as the one described as TAU. We have revised the sentence to make this clear: “Patients who did not wish to participate in this study were offered treatment corresponding to TAU”.

Interventions:

These remarks are connected to my other remarks about the difference between BBAT, especially its specific therapeutic goals, and mixed physical activity.

Therefore, I would recommend to describe the two arms separately and not under the common denominator of physical activity. By this subdivision the content of the manuscript will be more according to what is suggested in the title of the manuscript.

#2.12 Response: By now being more explicit in our definition of physical activity as well as mixed physical activity (please see #2.3) and regarding possible working mechanisms we have tried to avoid any confusion that the lack of clear definitions had previously brought to the manuscript. According to the rather broad definition of physical activity by Caspersen et al. both BBAT and the “mixed physical activity” fit in the definition.

Results

2.13: Lines 244/245

It seems rather obvious that, given a total time of ten months and an average attendance of ten sessions, so physiotherapy once a month, no effect is to be expected. I think this point should get more attention in the discussion. A heightened frequency and more focus on attendance (attendance being a general problem, see also the literature on physical activity for people with schizophrenia for example) could be a recommendation for the future.

#2.13 Response: This is an important comment. The decision of one weekly session was taken to be able to include as many as possible of the patients that are referred to the clinic to obtain an unselected sample. Many of the attending patients at CTP do not wish to come more than once weekly for a variety of reasons, including the cost of transport from their home to CTP which is not reimbursed. Knowing that frequency could not be more than once a week, we stressed the importance of home exercises, but due to a lack of systematic registering of home exercises we could not analyze if those that were persistent in doing exercises at home had a better outcome. As described in Methods PA was offered weekly from phase 1 and was thus not spread out over ten months, i.e. the frequency was not 1/months but 1/week in the period when PA was offered. from Lastly, the issue of low attendance also calls for attention. We have now carefully included this comment throughout the Discussion. We have also added to the Discussion that future studies could look at interventions designed to enhance motivation for physical activity as shown useful in studies on exercise in patients with schizophrenia spectrum disorder. We have added a relevant reference. 

That said it does not seem that the frequency alone accounts for the lack for difference between the physiotherapy groups and the TAU group since the per protocol (PP) analyses did not show any in difference in between groups neither. This is now further outlined in the Discussion section.

2.14: How many sessions were there for the intervention groups? I read 20 weekly sessions starting in phase one (line 159). Does this mean a total of 20 sessions?

#2.14 Response: Yes, this is correct. Both the mixed physical activity group and the BBAT group were offered 20 sessions each. However, due to non-attendance, as described few participants completed 20 sessions. To clarify this we have now added “offered” to the sentence so that it now reads: “The intervention groups B and M were offered 20 weekly sessions with a physiotherapist starting in phase 1”.

2.15: Line 250: was the percentage of comorbid depression established by the medical doctor according to ICD 10 criteria compared to the Hamilton depression scores? This seems sensible since the comorbid depression is surprisingly high (even given the fact that depression is the main comorbid disorder in PTSD).

#2.15 Response: We are aware of the very high prevalence of comorbid depression and believe that this illustrates the severe mental health problems in the sample. The high prevalence is similar to other randomized studies carried out at CTP. The comorbid depression was established by the medical doctor according to ICD-10 criteria. We have now specified this in the Participants section: “Based on pre-treatment interview with a medical doctor, PTSD, depression and personality change after catastrophic event were diagnosed according to a diagnostic algorithm following ICD-10 criteria.”

As explained in Randomisation and masking, all Hamilton Depression Ratings were carried out separately by blinded raters. The results of the Hamilton Depression Ratings also reflect a high level of depressive symptoms. The following sentence has been added to the section: “The clinicians were not given access to the results of the Hamilton interviews.” 

To underline the high comorbidity with depression we have added the following sentence in the last part of Strength and limitations: “The previous studies all have similar severity of PTSD at baseline, very high depression comorbidity (>90%) as well as a similar length of time in the new country.”

2.16: Table 2: The fitness results and the MAIA results are absent in this table, while they are crucial for the measurement of the add-on effects. So, it is not clear which measures are for what reason in the Supporting information.

#2.16 Response: With many outcomes we had to choose which ones to include in the main paper and which ones are included in the supplementary material. All tables will be available to all readers. We chose to include our CTP standard ratings in the main document as we expected to find add-on effects on these ratings.

If the editors disagree with this decision, we can of course revise the manuscript accordingly.

Discussion

2.17: Lines 315-316: I would suggest to delete the remark about the interventions in this study not being different from those in the Rosenbaum review. Interventions as well as populations are all very different in the Rosenbaum review, and are also different from the ones in the present study.

#2.17 Response: We have deleted the sentence stating that the interventions in the studies included in the reviews do not differ substantially from the intervention in the present study. We now have added instead: “The total number of sessions with PA in the studies included in the review do not differ substantially from the total number of sessions in our study.”

2.18: 345: duration is long, but frequency is a weakness!

#2.18 Response: We have now included this in our Discussion. (see also Response #2.14 above). 

2.19: 348: frequency of once a week would suggest 10 months = possibly 40 sessions!

#2.19 Response: We have now underlined that there were a maximum of 20 weekly sessions in the Strengths and limitations section (although also mentioned in the section The physical activity intervention). Please see also #2.14 above. 

2.20: 356-358 I would suggest to skip these lines and possibly look for other references. The authors rely too much on Rosenbaum et al.

#2.20 Response: As the review includes studies with 10-12 sessions with a similar frequency as the present we do believe that the review is useful for a comparison as the populations included are very different and we believe this to contribute to the differences found. However, we have modified the text: “The results from the present study seem to differ from the conclusion of a review including 4 studies of PA as adjunctive treatment for non-refugee populations with PTSD [9], which points at an improved treatment effect when using PA as adjunctive treatment”.

2.21: Lines 368-371: the add-ons are different because they use body- and movement interventions and it is strange to suggest that the bodily aspects were already met in the TAU.

#2.21 Response: We agree with this comment and believe that the sentences can easily be misunderstood. We have now deleted the specific sentences as the point of a chronic population with a potential limited window for improvement is already made in the manuscript.

2.22: Lines 374 and further: I have the impression the authors are disappointed, which I can imagine, but I think it would be worthwhile to think about other options, such as frequency of the add-on treatment, motivational issues, etc.

#2.22 Response: We have now in our discussion put more weight into discussing frequency, motivation and also chronicity. Please also see Response #2.13. The Discussion has been extensively revised.

Impact on future treatment.

2.23: The first lines should really be deleted. Adding physical activity 10 times in a ten month treatment and having no effect, does not justify this conclusion.

#2.23 Response: We do not believe that there should not be physical activity as a part of treatment offered selected groups of trauma-affected refugees. However, we do believe that the results from the present study should be taken into consideration when planning treatment in similar patient groups with as severe and chronic symptoms. There is a need for more knowledge on who should be offered PA as part of the treatment. This is now stressed, and we have adjusted the sentence regarding treatment planning and treatment guidelines to underline that it implies this patient group. 

Reviewer #3 

Present article describes a three-arm RCT to find the efficacy of physical activity as an added component in reducing symptoms of PTSD. Trial recruited 338 patients though there was a significant dropout in two arms. The description of trial is very clear with self-administered HTQ as the primary outcome. The study fall shorts in many other statistical and trial consideration, which I describe bellow.

3.1: Why the trial is “pragmatic” is not clear. Please describe it why or take the word out.

#3.1 Response: Thank you for pointing this out. A pragmatic RCT mimics usual clinical practice aiming at that results should be as applicable as possible to “real-world health settings”. Important criteria for a pragmatic trial are broad inclusion and having few exclusion criteria, which both were the case in the present trial. This has now been clarified in the last part of “Participants”: “The pragmatic design of this study primarily consisted of broad inclusion of the target group, only few exclusion criteria and an manualised flexible intervention offered at an outpatient clinic.”

3.2: How many data points per subject is collected on the course of trial is not clear.

#3.2 Response: In this paper only two data points were used: pre- and post-treatment. As stated in page 9, data was also collected for the self-ratings just before phase 2 of treatment. Further as stated at the end of the “Outcomes” section a follow up was carried out six months after ending the treatment programme. The results from the follow-up will be reported elsewhere. To avoid confusion and make the two data points clear, we have made some changes in the order and wording in the Outcomes section. In this paper, only pre- and post-treatment measurements will be analysed, i.e two data points.”

3.3: The reason for conducting three-arm superiority trial is not very clear. Note, inclusion of more number of arms increases the problem related to multiple testing. This has effect on sample size and type-1 error which I will describe next.

#3.3 Response: This remark is related to #2.9. We have revised the Introduction and among other things added the following: “The two categories of PA have different content and methodology, and different theories of working mechanisms, although overlap exists. Moreover, the focus on body awareness in mind-body interventions has obtained special attention in the PTSD literature due to theories of body awareness being a central aspect of the treatment of PTSD. Due to the possible different working mechanisms between structured exercise and mind-body interventions in the treatment of PTSD, in this study, we were interested both in studying if adding PA to treatment as usual (TAU) would increase treatment outcome but also if one of these two categories of PA was better than the other.”

3.4: The power analysis described in line 211-221 is insufficient. Please report standardized effects size (Cohen’s d) as it is not clear what does mean by 0.5 SD or 1SD. If I interpret 0.5SD and as Cohen’s d=0.5 then it is a moderately large effect size, but in the acceptable range. 1.0 is too large an effect size. It should also describe what statistical test is used and using what statistical model. It is not clear if a main effect is powered or an interaction term!! Since more than two arm is present there is a possibility of two or more co-primary hypothesis, which will require adjustment for multiplicity as otherwise type-1 error will be inflated (>0.05). This may render sample projected to be inferior. Primary analysis model and the model used for sample size should be same or closely following each other.

#3.4 Response: Cohen’s d is calculated as the group difference divided by the standard deviation, and consequently a d of 0.5 means a group difference corresponding to 0.5 standard deviation – in other words power calculation for “ ½ SD” is the same as power calculation for Cohen’s d = 0.5. We have made this point explicit in the revised manuscript. We have also made explicit that the power was calculated based on simple comparison of two independent groups, corresponding to post-treatment group differences. We did not adjust the power calculation for multiple testing, because we first conducted an overall test of significant mean differences between the three groups. It can be argued that the initial power calculation should correspond the statistical analysis (e. g. one-way ANOVA with three groups), but we find that Cohen’s d for differences between two groups are easier to interpret and it should also be remembered that the purpose of the power-calculation was to obtain a reasonable estimate of the number of patients to include in the study.

3.5: Also please mention which software or simulation method is used to do the power analysis.

#3.5 Response: The manuscript reports that all analyses were conducted in Stata. Since procedures for power calculation are now included in major statistical programs, we do not find it necessary to specifically mention that Stata was also used for power analyses. Should the editor disagree, we can of course include this information.

3.6: In line 225-228 many analysis methods are described, but I only see the result of pre-post type comparisons in the table 1 and table 2. Where is time by intervention result reported? Also line 234 is confusing as the result following the line is not reported. There is no table related to mixed-model ANOVA. If the study includes more than 3 observations per subject the mixed-effect model based ANOVA should be the primary result for determining superiority.

#3.6 Response: The manuscript mentions that “Table 2 illustrates the mixed model analyses on the ITT-sample”, and in fact all estimates and p-values in the table are based on the mixed model (the table headline also says that the results are based on the intention-to-treat population). Thus, all means and SEs are mixed model estimates and so are the p-values. The p-values in pre- and post-treatment columns refer to an overall test of the significance mean differences between the three groups (corresponding to an ANOVA F-test). The p-values in the column showing pre-post treatment differences refer to an overall test of the significance of group differences in mean pre-post treatment differences (this test of group differences in pre-post treatment difference corresponds to the statistical test of the time by intervention interaction). Finally, the column with p-values shows p-values corresponding to the pre-post treatment mean difference in each of the three groups.

We find that this table is a compact and informative way of presenting the results of the mixed model analyses and have revised the table foot note to make this clearer to the reader.

3.7: ITT and per-protocol analysis both are done and it is not clear which result should be more important. It is sometime confusing and not clear reasoning is mentioned for both.

#3.7 Response: The results of the ITT analysis are presented in the manuscript while the results of the per-protocol analysis are presented in an online table. This should clarify the issue, but we have gone through the manuscript to check for any ambiguities and now it explicitly describes how the per-protocol analysis may supplement the ITT analysis. In the Statistical analysis section we have specified that ITT is the primary analysis: “This analysis was carried out both as the primary intention-to-treat analyses of all participants who completed pre-treatment ratings, and in addition on a reduced sample (per-protocol analyses)”.

3.8: A major concern near 50% dropout in Arm B and M as presented in Figure 1. From the reason for dropout it seems the dropout are non-ignorable as a result if proper care is not taken the results are prone to selection bias. Neither ITT not PPA can take care of this high attrition. A proper approach will be modelling the Missing Not at Random (MNAR) mechanism. I am really doubtful if this aspect is taken care of.

#3.8 Response: This is a trial carried out in a “real-life” mental health setting which is reflected in the attendance. However, we do understand the concern of the reviewer. We are though not completely sure if the reviewer might have misunderstood Figure 1. There were 77 who were lost to follow up (n=77 C/B/M 23/22/32) and 20 withdrawn from the study. However, the numbers mentioned above with 50% concern number of completers in Arm B and M. As described in the last section of Statistical analysis. “The per-protocol population was defined as all patients in the control group plus all from B and M who participated in ≥10 physiotherapy sessions.” 

Thus, we defined completers as those having attended a minimum of ten sessions with a physiotherapist among those offered physiotherapy. Therefore, there could by definition only be completers in the TAU. It is of course worrying that so few actually carried through more than ten sessions with the physiotherapist as to expected outcome and this is now more extensively discussed in the Discussion (see response #2.13 above).

---

## [Decision Letter · Decision Letter 1]

13 Dec 2019

PONE-D-19-16080R1

Trauma-affected refugees treated with basic body awareness therapy or mixed physical activity as augmentation to treatment as usual – a pragmatic randomised controlled trial

PLOS ONE

Dear Jessica Carlsson Carlsson,

Thank you for submitting your manuscript to PLOS ONE. After careful consideration, we feel that it has merit but does not fully meet PLOS ONE’s publication criteria as it currently stands. Therefore, we invite you to submit a revised version of the manuscript that addresses the points raised during the review process.

Albeit the manuscript presents improvements from the orginal version, there are still some substantial point, dtailed in the reviewers' reports, that requre attention and proper action. 

We would appreciate receiving your revised manuscript by January 31. To enhance the reproducibility of your results, we recommend that if applicable you deposit your laboratory protocols in protocols.io, where a protocol can be assigned its own identifier (DOI) such that it can be cited independently in the future. For instructions see: http://journals.plos.org/plosone/s/submission-guidelines#loc-laboratory-protocols

We look forward to receiving your revised manuscript.

Kind regards,

Andrea Martinuzzi

Academic Editor

PLOS ONE

Reviewers' comments:

Reviewer's Responses to Questions

**Comments to the Author**

1. If the authors have adequately addressed your comments raised in a previous round of review and you feel that this manuscript is now acceptable for publication, you may indicate that here to bypass the “Comments to the Author” section, enter your conflict of interest statement in the “Confidential to Editor” section, and submit your "Accept" recommendation.

Reviewer #1: All comments have been addressed

Reviewer #2: (No Response)

Reviewer #4: (No Response)

2. Is the manuscript technically sound, and do the data support the conclusions?

Reviewer #1: Yes

Reviewer #2: Yes

Reviewer #4: Partly

3. Has the statistical analysis been performed appropriately and rigorously? 

Reviewer #1: Yes

Reviewer #2: Yes

Reviewer #4: No

4. Have the authors made all data underlying the findings in their manuscript fully available?

Reviewer #1: Yes

Reviewer #2: Yes

Reviewer #4: Yes

5. Is the manuscript presented in an intelligible fashion and written in standard English?

Reviewer #1: Yes

Reviewer #2: Yes

Reviewer #4: Yes

6. Review Comments to the Author

Reviewer #1: As I said in the first review, the research findings did not support the idea of adding physical activity programme to TAU for refugees affected by PTSD. It appears not important to define the physical activity the authors are talking about.

Reviewer #2: The manuscript has been largely improved! It has been made clear why two different categories of physical activity have been chosen. Also, the differences between structured exercise and mind-body interventions have been outlined in more detail. Next, other issues I raised in my comments have been addressed satisfactorily. Below I state some remaining, mostly small, questions and comments.

General remark: The manuscript still contains several typos and double spaces. I addressed some below, but I advise to re-read the manuscript.

Abstract:

Lines 41-44

The authors now state: ‘A large number of the participants in the study have a chronic mental condition, often difficult to treat.’

However, they do not explain the link between the lack of effect related to PA and the chronic mental condition. So, this phrase does not add much in terms of explanations or hypotheses.

Line 54

‘postmigration’ should be post-migration

Line 65

‘andthe suggests’ should be corrected

Line 67

mind body interventions should be mind-body interventions

Lines 80-81

‘Physical activity in this study, was defined as suggested by Caspersen et al. “any bodily movement produced by skeletal muscles that results in energy expenditure ” [13]. ‘

I think after Caspersen et al. the word ‘as’ is missing. Furthermore, because this is a quotation, a page number should be added.

‘Further’ PA can be divided in etc. should be Furthermore,

Lines 151/152

‘Patients who did not wish to participate in this study were offered treatment corresponding to TAU’

I would say: ‘similar’ to TAU, but I am not a native speaker

Line 211

(e.g.. body scan, rotation, the wave).

I suggest to delete these examples, because for the lay person, they do not provide an explanation and only rise more questions. May be apart from the body scan, but a body scan is not a typical BBAT exercise.

Line 216

everyday-like tool should be tools

Line 249

‘ i.e two data points’.

I think this information is reduntant.

Line 366

‘primary and on many secondary outcomes’

Many secondary outcomes is rather vague.

Line 370

4 studies should be four studies

Lines 380-381

‘Several studies have previously suggested that treatment effects in

trauma-affected refugees are typically smaller than in other trauma-affected populations[24-26].’

These studies have been conducted by some of the present researchers. Therefore, if available, It would be good to add another reference, from another country/group of researchers.

Lines 405-406

‘By using individual treatment sessions, no potential spillover effect of a group could blur the results.’

Did the authors also consider the possibility that a groupwise treatment could enhance the effects?

I would like to point out that the motivational intervention cited by the authors in line 436 (reference 46) is also a group intervention !

Line 408

lack for difference should be lack of difference

Reviewer #4: The manuscript entitled 'Trauma-affected refugees treated with basic body awareness therapy or mixed physical activity as augmentation to treatment as usual – a pragmatic randomised controlled trial' with the aim to investigate if adding either BBAT or mixed physical activity to the treatment as usual (TAU) for trauma-affected refugees with PTSD would increase the treatment effect compared to TAU alone.

Comments

Questionnaires

The language version that was used by the patients to be cited and referenced for each questionnaire.

For the baseline characteristics, the breakdown of education level to be provided. A statement on whether they are able to read/write and understand the questionnaire to be stated.

The results presentations should place more emphasis on the comparison of pre and post treatment of each group rather than focusing on the group difference at pre treatment and post treatment respectively altogether. Even though there was no significant different at baseline between the groups, the figures are still different from each other. This may affect on the post treatment value. Also based on CONSORT guidelines, all baseline comparison to be avoided.

Page 11 Line 227, for the sentence ' pre treatment interview, shortly before initiating phase two and at the last treatment' the word post treatment to be used.

Page 11 Line 244-246, the outcome variable used for the sample size calculation to be stated. Other information to be provided e.g. alpha, mean, which two groups were used, software, formula etc. Post hoc multiple comparison method to be considered in the sample size calculation. There was no information on the attrition rates if it were taken into consideration of the sample size calculation.

Page 12 Line 250 - 251, the sentences to be placed after Line 264.

Page 12 Line 253, 1 tailed or 2 tailed test to be stated. Fishers exact test to be rewritten as Fisher's exact test. The level of acceptance significance to be stated.

Page 13 Table 1, at least one decimal point for percentages to be provided. The symbol % to be omitted since it has been indicated on the top. w.loss to be spelled out. F.62.0 to be denoted in the table footnote. The sentence 'There were no significant or borderline significant group differences among the data above' to be break up and denoted in table footnote. Symbol N for All, group C, B and M on to[ of table.

Table 2, effect size could be explored. Symbol <= to be replaced with ≤. Statistical test to be denoted in the table footnote. Decimal points for p value to be standardized.

S1 Table & S2 Table, few figures with 1 decimal point (to be consistent with 2 decimal points).

References list did not conform to the journal format.

7. PLOS authors have the option to publish the peer review history of their article (what does this mean?). If published, this will include your full peer review and any attached files.

Reviewer #1: No

Reviewer #2: Yes: Mia Scheffers

Reviewer #4: No

---

## [Author Response · Author response to Decision Letter 1]

27 Jan 2020

PONE-D-19-16080

Trauma-affected refugees treated with basic body awareness therapy or mixed physical activity as augmentation to treatment as usual – a pragmatic randomised controlled trial

PLOS ONE

Dear Editor and reviewers

Thank you for offering the opportunity to further improve the manuscript based on your comments

Review Comments to the Author (after first revision)

Reviewer #1: As I said in the first review, the research findings did not support the idea of adding physical activity programme to TAU for refugees affected by PTSD. It appears not important to define the physical activity the authors are talking about.

1. Response: Adding a definition of physical activity was a request by Reviewer#2 and therefore kept in the manuscript.

Reviewer #2: The manuscript has been largely improved! It has been made clear why two different categories of physical activity have been chosen. Also, the differences between structured exercise and mind-body interventions have been outlined in more detail. Next, other issues I raised in my comments have been addressed satisfactorily. Below I state some remaining, mostly small, questions and comments.

2.1 General remark: The manuscript still contains several typos and double spaces. I addressed some below, but I advise to re-read the manuscript.

2.1 Response: The manuscript has been carefully re-read to detect typos and double spaces.

2.2 Abstract:

Lines 41-44

The authors now state: ‘A large number of the participants in the study have a chronic mental condition, often difficult to treat.’

However, they do not explain the link between the lack of effect related to PA and the chronic mental condition. So, this phrase does not add much in terms of explanations or hypotheses.

2.2 Response: We do agree that as it stands this sentence does not clearly explain the link. However, due to a limitation in number of words in the abstract (word count 309 with a limit of 300) and as the question of chronicity is elaborated in the Discussion we have now chosen to delete the sentence in the abstract. 

2.3 Line 54

‘postmigration’ should be post-migration

2.3 Response: This has been corrected.

2.4 Line 65

‘andthe suggests’ should be corrected

2.4 Response: “the” has been deleted.

2.5 Line 67

mind body interventions should be mind-body interventions

2.5 Response: This has been corrected.

2.6 Lines 80-81

‘Physical activity in this study, was defined as suggested by Caspersen et al. “any bodily movement produced by skeletal muscles that results in energy expenditure ” [13]. ‘

I think after Caspersen et al. the word ‘as’ is missing. Furthermore, because this is a quotation, a page number should be added.

2.6 Response: The word “as” has now been added. The page number has been added just after the in-text citation. 

2.7 ‘Further’ PA can be divided in etc. should be Furthermore

2.7 Response: Further has been changed to Furthermore.

2.8 Lines 151/152

‘Patients who did not wish to participate in this study were offered treatment corresponding to TAU’

I would say: ‘similar’ to TAU, but I am not a native speaker

2.8 Response: The sentence has now been changed to: “Patients who did not wish to participate in this study were offered treatment as described in the TAU manual”.

2.9 Line 211

(e.g.. body scan, rotation, the wave).

I suggest to delete these examples, because for the lay person, they do not provide an explanation and only rise more questions. May be apart from the body scan, but a body scan is not a typical BBAT exercise.

2.9 Response: To avoid confusion we have now deleted “e.g.. body scan, rotation, the wave” as suggested.

2.10 Line 216

everyday-like tool should be tools

2.10 Response: Everyday-like tools has been changed to tools.

2.11 Line 249

‘ i.e two data points’.

I think this information is reduntant.

2.11 Response: “i.e two data points” was added as a response to a question regarding number of data points made by Reviewer #3 in the first review and is therefore kept in this revision.

2.12 Line 366

‘primary and on many secondary outcomes’

Many secondary outcomes is rather vague.

2.12 Response: We agree and “many” has now been changed to the specific number (four) and the names of the outcomes have been added. 

“…on the primary and on four of the secondary outcomes (WHO, GAF-S, HoNOS, MAIA -not worrying)…”

2.13 Line 370

4 studies should be four studies

2.13 Response: This has been corrected to four studies.

2.14 Lines 380-381

‘Several studies have previously suggested that treatment effects in

trauma-affected refugees are typically smaller than in other trauma-affected populations[24-26].’

These studies have been conducted by some of the present researchers. Therefore, if available, It would be good to add another reference, from another country/group of researchers.

2.14 Response: As requested we have added a reference from a separate group of researchers (Ter Heide et al., Eye movement desensitisation and reprocessing therapy v. stabilisation as usual for refugees: randomised controlled trial, Br J Psychiatry 2016).

2.15 Lines 405-406

‘By using individual treatment sessions, no potential spillover effect of a group could blur the results.’

Did the authors also consider the possibility that a groupwise treatment could enhance the effects?

I would like to point out that the motivational intervention cited by the authors in line 436 (reference 46) is also a group intervention !

2.15 Response: We did consider the possible advantages of group sessions. However, there was a main reason for not choosing group treatment which is now explained: “Additionally, BBAT as method was prior to this study tested in a pilot study on the same target group, and the method was found satisfactory and acceptable. In the pilot study, BBAT was offered in group sessions, which was the reason for several patients not wishing to participate in the study. In the present study we therefore avoided group sessions as this had been reported to be an obstacle for participation.”

2.16 Line 408

lack for difference should be lack of difference

2.16 Response: This has been corrected.

Reviewer #4: The manuscript entitled 'Trauma-affected refugees treated with basic body awareness therapy or mixed physical activity as augmentation to treatment as usual – a pragmatic randomised controlled trial' with the aim to investigate if adding either BBAT or mixed physical activity to the treatment as usual (TAU) for trauma-affected refugees with PTSD would increase the treatment effect compared to TAU alone.

Comments

4.1 Questionnaires

The language version that was used by the patients to be cited and referenced for each questionnaire.

4.1 Response: As stated in page 11: “All self-administered outcomes were available in the five main languages of the patients (Danish, Arabic, English, Bosnian/Serbo-Croatian and Farsi).”. There are only few references for the different language versions of the included questionnaires. These have now been included in the section Outcomes (Methods). 

4.2 For the baseline characteristics, the breakdown of education level to be provided. A statement on whether they are able to read/write and understand the questionnaire to be stated.

4.2 Response: In the data material the education variable contains information on number of years of education in the home country (<2 years, 2-5 years, >5-10 years, >10-15 years and more than >15 years). This more detailed presentation of data on education has been inserted in table 1. 

Most participants were able to read and fill in the questionnaires themselves. “There were a few illiterate (number unknown) participants and also a few participants where the questionnaires had not been translated to their language. These participants were assisted by an interpreter.” This has been added in Outcomes (Methods), page 11. We have also added a sentence regarding assistance by interpreter during treatment at the end of the section Participants (Methods), page 8: “All patients in need of an interpreter received this assistance and if possible, the same interpreter was used throughout the treatment.”

4.3 The results presentations should place more emphasis on the comparison of pre and post treatment of each group rather than focusing on the group difference at pre treatment and post treatment respectively altogether. Even though there was no significant different at baseline between the groups, the figures are still different from each other. This may affect on the post treatment value. 

4.3 Response: We find that the current table 2 presents all relevant means and pre-post treatment differences. Even though post-treatment group differences are often used as the primary outcome in randomised trials, pre-treatment group differences might affect the post-treatment group differences in absolute scores, and even though p-values for these differences are provided, the reviewer is right to stress the pre-post treatment differences in each group and in particular, the group differences in pre-post treatment changes. The latter group differences are likely to be little affected by pre-treatment differences and are an important test of different effects of the three treatment interventions. 

As an alternative we have also analysed the data using a regression model with indicator variables for the three groups and the pre-treatment score as independent variable and the post-treatment score as outcome variable. This analysis – which fully adjusted for pre-treatment differences among the three groups – showed no significant group differences in post-treatment means, confirming the results of the mixed model analysis.

4.4 Also based on CONSORT guidelines, all baseline comparison to be avoided.

4.4 Response: We have now deleted the following sentence regarding baseline comparisons: “No significant group differences were seen at baseline on any of the characteristics listed in Table 1.” We have furthermore (for the same reason) deleted the sentence: 'There were no significant or borderline significant group differences among the data above” in table 1. 

4.5 Page 11 Line 227, for the sentence ' pre treatment interview, shortly before initiating phase two and at the last treatment' the word post treatment to be used.

4.5 Response: The word post-treatment is now used as suggested.

4.6 Page 11 Line 244-246, the outcome variable used for the sample size calculation to be stated. Other information to be provided e.g. alpha, mean, which two groups were used, software, formula etc. Post hoc multiple comparison method to be considered in the sample size calculation. There was no information on the attrition rates if it were taken into consideration of the sample size calculation.

4.6 Response: The power calculations were conducted with regard to the HTQ, and this has been made more clear in the revised version of the paragraph where we also include the alpha level and a comment on multiple comparison – although due to lack of significant overall effects comparison of means was not used for the main outcomes. Concerning attrition rates the manuscript makes clear that power should be sufficient with about 200 participants, and that the study – because of expected attrition - was planned to include 250 participants, but that 338 patients were in fact included.

The revised paragraph on power now reads: 

“Power calculations were conducted with regard to the primary outcome HTQ. The study was initially planned to include approximately 250 patients as a conservative estimate of 200 participants eligible for intention-to-treat analysis with about 65 in each of the three groups would provide power of 81% to detect a group difference corresponding to a standardised difference of ½ SD (Cohen’s d) and power to detect a difference of 1 SD of close to 100% (using a 5% level of significance and planning to use contrasts to compare group means in case of a significant overall test of group differences). A difference of less than ½ SD was considered less relevant from a clinical perspective. However, due to a smaller number than expected of male patients with a high HTQ score (which we stratified for in accordance with the average level of HTQ-score in previous RCTs conducted at CTP [24,25]) a larger number was included. Thus, the final number of included patients was 338.”

4.7 Page 12 Line 250 - 251, the sentences to be placed after Line 264.

4.7 Response: These two sentences have been moved to the end of “Statistical analysis” as suggested.

4.8 Page 12 Line 253, 1 tailed or 2 tailed test to be stated. Fishers exact test to be rewritten as Fisher's exact test. The level of acceptance significance to be stated.

4.8 Response: In the statistical paragraph the following sentence has been included in the revised version of the manuscript: Two-tailed tests with a 5% level of significance were used in all statistical tests. In addition, the spelling of Fisher’s exact test has been corrected

4.9 Page 13 Table 1, at least one decimal point for percentages to be provided. The symbol % to be omitted since it has been indicated on the top. w.loss to be spelled out. F.62.0 to be denoted in the table footnote. The sentence 'There were no significant or borderline significant group differences among the data above' to be break up and denoted in table footnote. Symbol N for All, group C, B and M on to[ of table.

4.9 Response: 

• One decimal is now provided for all the percentages

• The symbol % has been taken out

• “Cranial trauma with loss of consciousness” is now spelled out.

• The code F62.0 has been deleted from the table as the table does not contain other codes for diagnoses.

• The sentence on significant group differences is taken out (please see response 4.4)

• Symbol N is now inserted for All, group C, B and M in the top of the table.

4.10 Table 2, effect size could be explored. Symbol <= to be replaced with ≤. Statistical test to be denoted in the table footnote. Decimal points for p value to be standardized.

4.10 Response: 

• Effect size: Since table 2 does not show a single significant group difference in absolute post-treatment scores or in changes between pre- and post-treatment scores, it hardly makes sense to calculate effect size. However, several measures show significant changes from pre- to post-treatment scores. Since there are no significant group differences in these changes, it makes sense to calculate estimates of effect size based on the full sample. The range of these effect sizes vary from 0.07 (BPI interference) to 0.57 (GAF-S), using the baseline SD to standardise the changes from pre- to post-treatment. Corresponding to the levels of significance in the table, the largest estimated effect sizes were 0.48 SD (HTQ), 0.53 SD (WHO-5) and 0.57 SD (GAF-S). Since these changes are unrelated to the investigated interventions, they may reflect effects of TAU or non-treatment related factors. Because the focus of the paper is on the effects of the interventions, we have not found it appropriate to discuss these effect estimates in the revised paper, but this can be included if the editor finds that it should.

• The symbol <= has been replaced with ≤ in table 2, S1 and S2

• Denotation of statistical test: Manuscript table 2 includes the footnote below. We find that this is a very detailed explanation and see no need to expand this description. Should the editor disagree we are of course willing to do so.

“All means and SEs in table 2 are mixed model estimates and so are the p-values. The p-values in the pre- and post-treatment columns refer to an overall test of the significance mean differences between the three groups (corresponding to an ANOVA F-test). The p-values in the column showing pre-post treatment differences refer to an overall test of the significance of group differences in mean pre-post treatment differences (this test of group differences in pre-post treatment difference corresponds to the statistical test of the time by intervention interaction). Finally, the column with p-values shows p-values corresponding to the pre-post treatment mean difference in each of the three groups.”

• The decimal points for the p value have been standardised as requested.

4.11 S1 Table & S2 Table, few figures with 1 decimal point (to be consistent with 2 decimal points).

4.11 Response: S1 and S2 have now been carefully looked through and corrected as to inconsistency in number of decimals.

4.12 References list did not conform to the journal format.

4.12 Response: The reference list has been corrected to conform with the journal format.

---

## [Decision Letter · Decision Letter 2]

27 Feb 2020

Trauma-affected refugees treated with basic body awareness therapy or mixed physical activity as augmentation to treatment as usual – a pragmatic randomised controlled trial

PONE-D-19-16080R2

Dear Dr. Carlsson,

We are pleased to inform you that your manuscript has been judged scientifically suitable for publication and will be formally accepted for publication once it complies with all outstanding technical requirements.

With kind regards,

Andrea Martinuzzi

Academic Editor

PLOS ONE

Additional Editor Comments (optional):

Reviewers' comments:

Reviewer's Responses to Questions

**Comments to the Author**

1. If the authors have adequately addressed your comments raised in a previous round of review and you feel that this manuscript is now acceptable for publication, you may indicate that here to bypass the “Comments to the Author” section, enter your conflict of interest statement in the “Confidential to Editor” section, and submit your "Accept" recommendation.

Reviewer #2: All comments have been addressed

Reviewer #4: All comments have been addressed

2. Is the manuscript technically sound, and do the data support the conclusions?

Reviewer #2: Yes

Reviewer #4: (No Response)

3. Has the statistical analysis been performed appropriately and rigorously? 

Reviewer #2: Yes

Reviewer #4: (No Response)

4. Have the authors made all data underlying the findings in their manuscript fully available?

Reviewer #2: Yes

Reviewer #4: (No Response)

5. Is the manuscript presented in an intelligible fashion and written in standard English?

Reviewer #2: Yes

Reviewer #4: (No Response)

6. Review Comments to the Author

Reviewer #2: Thank you for this important contribution to the field! This study is important for all working with refugees with trauma-related problems!

Reviewer #4: (No Response)

7. PLOS authors have the option to publish the peer review history of their article (what does this mean?). If published, this will include your full peer review and any attached files.

Reviewer #2: Yes: Mia Scheffers

Reviewer #4: No

---

## [Editor Report · Acceptance letter]

2 Mar 2020

PONE-D-19-16080R2 

Trauma-affected refugees treated with basic body awareness therapy or mixed physical activity as augmentation to treatment as usual – a pragmatic randomised controlled trial 

Dear Dr. Carlsson:

I am pleased to inform you that your manuscript has been deemed suitable for publication in PLOS ONE. Congratulations! Your manuscript is now with our production department. 

With kind regards,

on behalf of

Dr. Andrea Martinuzzi 

Academic Editor

PLOS ONE